# The homo-oligomerisation of both Sas-6 and Ana2 is required for efficient centriole assembly in flies

**Matthew A Cottee†, Nadine Muschalik†‡, Steven Johnson, Joanna Leveson, Jordan W Raff*, Susan M Lea***

Sir William Dunn School of Pathology, University of Oxford, Oxford, United Kingdom

**Abstract** Sas-6 and Ana2/STIL proteins are required for centriole duplication and the homo-oligomerisation properties of Sas-6 help establish the ninefold symmetry of the central cartwheel that initiates centriole assembly. Ana2/STIL proteins are poorly conserved, but they all contain a predicted Central Coiled-Coil Domain (CCCD). Here we show that the *Drosophila* Ana2 CCCD forms a tetramer, and we solve its structure to 0.8 Å, revealing that it adopts an unusual parallel-coil topology. We also solve the structure of the *Drosophila* Sas-6 N-terminal domain to 2.9 Å revealing that it forms higher-order oligomers through canonical interactions. Point mutations that perturb Sas-6 or Ana2 homo-oligomerisation in vitro strongly perturb centriole assembly in vivo. Thus, efficient centriole duplication in flies requires the homo-oligomerisation of both Sas-6 and Ana2, and the Ana2 CCCD tetramer structure provides important information on how these proteins might cooperate to form a cartwheel structure.

**\*For correspondence:** jordan.raff@path.ox.ac.uk (JWR); susan.lea@path.ox.ac.uk (SML)

†These authors contributed equally to this work

**Present address:** ‡Division of Cell Biology, MRC-Laboratory of Molecular Biology, Cambridge, United Kingdom

**Competing interests:** The authors declare that no competing interests exist.

## Introduction

Centrioles are complex microtubule (MT) based structures that are required for the formation of centrosomes and cilia/flagella. These organelles have many important functions in cells, and their dysfunction has been linked to a plethora of human pathologies, ranging from cancer to microcephaly to obesity (*Nigg and Raff, 2009*; *Bettencourt-Dias et al., 2011*). Thus, understanding how these organelles assemble and function is an important goal of both basic and biomedical research.

Although several hundred proteins are thought to be concentrated at centrioles, only a small number appear to form a conserved 'core' pathway that is essential for centriole assembly (*Delattre et al., 2006*; *Pelletier et al., 2006*; *Gönczy, 2012*). During canonical centriole duplication, the protein kinase Plk4/Sak/ZYG-1 is recruited to the mother centriole by SPD-2 in worms (*Delattre et al., 2006*; *Pelletier et al., 2006*; *Shimanovskaya et al., 2014*), by Asterless (Asl) in flies (*Blachon et al., 2008*; *Dzhindzhev et al., 2010*), or by a combination of the two (Cep192 and Cep152, respectively) in humans (*Cizmecioglu et al., 2010*; *Hatch et al., 2010*; *Kim et al., 2013*; *Sonnen et al., 2013*). The protein kinase recruits STIL/Ana2/SAS-5 and Sas-6 to a single site on the side of the mother centriole where they assemble with CPAP/Sas-4 into a cartwheel structure that helps to establish the ninefold symmetry of the centriole (*Dammermann et al., 2004*; *Delattre et al., 2004*; *Leidel et al., 2005*; *Nakazawa et al., 2007*; *Peel et al., 2007*; *Strnad et al., 2007*; *Stevens et al., 2010a*; *Tang et al., 2011*; *Arquint et al., 2012*). CPAP/Sas-4 can interact with tubulin (*Hung et al., 2004*) and is required to recruit the centriole MTs to the outer region of the cartwheel (*Pelletier et al., 2006*), possibly working together with Cep135/Bld10 (*Hiraki et al., 2007*; *Lin et al., 2013*)—although no homologue of this protein has been identified in worms, and it does not appear to be essential for centriole duplication in flies (*Carvalho-Santos et al., 2012*; *Mottier-Pavie and Megraw, 2009*; *Roque et al., 2012*).

**eLife digest** Most animal cells contain structures known as centrioles. Typically, a cell that is not dividing contains a pair of centrioles. But when a cell prepares to divide, the centrioles are duplicated. The two pairs of centrioles then organize the scaffolding that shares the genetic material equally between the newly formed cells at cell division.

Centriole assembly is tightly regulated and abnormalities in this process can lead to developmental defects and cancer. Centrioles likely contain several hundred proteins, but only a few of these are strictly needed for centriole assembly. New centrioles usually assemble from a cartwheel-like arrangement of proteins, which includes a protein called SAS-6. Previous work has suggested that in the fruit fly *Drosophila melanogaster*, Sas-6 can only form this cartwheel when another protein called Ana2 is also present, but the details of this process are unclear.

Now, Cottee, Muschalik et al. have investigated potential features in the Ana2 protein that might be important for centriole assembly. These experiments revealed that a region in the Ana2 protein, called the 'central coiled-coil domain', is required to target Ana2 to centrioles. Furthermore, purified coiled-coil domains were found to bind together in groups of four (called tetramers). Cottee, Muschalik et al. then used a technique called X-ray crystallography to work out the three-dimensional structure of one of these tetramers and part of the Sas-6 protein with a high level of detail. These structures confirmed that Sas-6 proteins also associate with each other.

When fruit flies were engineered to produce either Ana2 or Sas-6 proteins that cannot self-associate, the flies' cells were unable to efficiently make centrioles. Furthermore, an independent study by Rogala et al. found similar results for a protein that is related to Ana2: a protein called SAS-5 from the microscopic worm *Caenorhabditis elegans*.

Further work is needed to understand how Sas-6 and Ana2 work with each other to form the cartwheel-like arrangement at the core of centrioles.

Great progress has been made recently in understanding how these proteins interact and how these interactions are regulated to ensure that a new centriole is only formed at the right place and at the right time. In particular, the crystal structure of Sas-6 from several species has revealed how this protein forms a dimer through its C-terminal coiled-coil domain (C–C) that can then further homo-oligomerise through an N-terminal headgroup interaction (N–N) to form a ring structure from which the C–C domains emanate as spokes (*Kitagawa et al., 2011*; *van Breugel et al., 2011*, *2014*; *Hilbert et al., 2013*). This Sas-6 ring structure can be modelled into EM tomographic reconstructions of the cartwheel from *Trichonympha* centrioles (*Guichard et al., 2012*, *2013*), strongly suggesting that these Sas-6 rings form the basic building blocks of the cartwheel. In support of this hypothesis, mutant forms of Sas-6 that cannot homo-oligomerise through the N–N interaction are unable to support efficient centriole duplication (*Kitagawa et al., 2011*; *van Breugel et al., 2011*), although they can still target to centrioles, a function that seems to rely on the C–C domain (*Fong et al., 2014*; *Keller et al., 2014*).

A crystal structure of the interface between Ana2/STIL and Sas-4/CPAP has also recently been solved (*Cottee et al., 2013*; *Hatzopoulos et al., 2013*), as has the interaction interface between Plk4 and both Cep192/SPD-2 and Cep152/Asl (*Park et al., 2014*); mutations that perturb these interactions in vitro perturb centriole duplication in vivo, indicating that these interactions are also essential for centriole duplication. More recently, it has been shown that Plk4 can recruit STIL to centrioles in human cells (*Ohta et al., 2014*; *Kratz et al., 2015*) and that Plk4/Sak can phosphorylate the conserved **ST**IL/**An**a2 (STAN) domain in STIL/Ana2 proteins in humans and flies, thereby promoting the interaction of the STAN domain with Sas-6 (*Dzhindzhev et al., 2014*; *Ohta et al., 2014*; *Kratz et al., 2015*). Mutant forms of STIL/Ana2 that could not be phosphorylated strongly perturbed Sas-6 recruitment to centrioles and centriole duplication. Together, these studies have shed important light on the molecular mechanisms of centriole assembly, but many important questions remain.

In particular, it has been proposed that the homo-oligomerisation properties of Sas-6 establish the ninefold symmetry of the centriole (*Kitagawa et al., 2011*), and, remarkably, a ninefold symmetric ring structure is formed in crystallo by *Leishmania major* Sas-6 (*van Breugel et al., 2014*). However, although Sas-6 oligomers appear to have a propensity towards ninefold symmetry, Sas-6 proteins

spontaneously assemble into oligomers of varying stoichiometry in vitro (*Kitagawa et al., 2011*; *van Breugel et al., 2011*), suggesting that the homo-oligomerisation properties of Sas-6 alone may be insufficient to enforce the rigorous ninefold symmetry that is observed in centrioles from virtually all species (*Cottee et al., 2011*). Additionally, recent Cryo-EM analysis suggests that the basic building block of the cartwheel stack is not a single ring and spoke structure, but rather a pair of rings that sit on top of one another: these rings do not make direct contact with each other, but are joined in the more peripheral regions through their spokes (*Guichard et al., 2012*, *2013*). Our current knowledge of Sas-6 self-association cannot explain this important feature of the cartwheel structure.

We previously showed that overexpressed Sas-6 can form higher-order aggregates in *Drosophila* spermatocytes, but these aggregates only adopt a cartwheel-like structure when Ana2 is also overexpressed (*Stevens et al., 2010b*), and the STIL/Ana2 protein family is essential for the proper recruitment of Sas-6 to centrioles (*Dzhindzhev et al., 2014*; *Ohta et al., 2014*). We reasoned therefore, that Ana2 was likely to also play an important part in determining the structure of the central cartwheel. We set out to investigate the potential structural features of Ana2 that might be important for centriole assembly.

## Results

### The CCCD is required for the centriolar targeting of Ana2

The *Drosophila* Ana2 protein contains four regions that have significant homology to Ana2/STIL proteins from other species (*Figures 1A, 2A*) (*Cottee et al., 2013*). Fly Ana2 lacks the conserved region 1 found towards the N-terminus in vertebrate STIL proteins (*Figure 2A*), but contains a CR2 domain that interacts with Sas-4 (*Cottee et al., 2013*; *Hatzopoulos et al., 2013*), a predicted central coiled-coiled domain (CCCD), a STAN domain (*Stevens et al., 2010a*) that interacts with Sas-6 (*Dzhindzhev et al., 2014*; *Ohta et al., 2014*) and a short C-terminal CR4 domain (*Figure 1A*) (*Cottee et al., 2013*). To examine the potential function of these conserved regions, we synthesised mRNAs in vitro that contained either wild type (WT) or truncated versions of Ana2 fused to either an N- or C-terminal GFP (*Figure 1A*). These mRNAs were injected into WT early embryos (that contain unlabelled endogenous WT Ana2 protein) expressing RFP-Centrosomin (Cnn) as a centrosomal marker (*Conduit et al., 2010*). The localisation of the encoded GFP-fusion protein was assessed 90–120 min after mRNA injection (*Figure 1B,C*).

Both N- and C-terminal GFP fusions of full length Ana2 (constructs 1 and 6) showed a strong, compact, localisation to centrioles, as did fusions lacking either CR2 or CR4 (constructs 2 and 7), suggesting that these domains are not involved in Ana2 centriolar targeting. In contrast, fusions retaining the STAN domain, but lacking the CCCD (constructs 8, 9 and 11) showed a weak and diffuse localisation to the PCM. This PCM localisation appeared to be dependent on the STAN domain, as constructs lacking both the CCCD and the STAN domain were no longer detectable at centrioles or in the PCM (constructs 5, 10, 12 and 13). In contrast, constructs lacking the STAN domain, but retaining the CCCD, localised as a tight dot to centrioles (although much more weakly than constructs that contained both domains) and were not detectable in the PCM (constructs 3 and 4). These observations suggest that the CCCD is required for the centriolar localisation of Ana2, while the STAN domain increases the efficiency of centriolar localisation and can also weakly target Ana2 to the PCM if the CCCD is absent. These findings are in agreement with recent data showing that STIL, the human homologue of Ana2, is recruited to centrioles through a direct interaction between regions of STIL containing the CCCD and Plk4 (*Ohta et al., 2014*; *Kratz et al., 2015*). Interestingly, the CCCD alone could not target GFP to centrioles (constructs 14 and 15), demonstrating that, at least in this context, the CCCD was not able to directly target proteins to the centriole.

### The CCCD forms a stable tetramer in solution

We reasoned that the CCCD might function as an oligomerisation domain for Ana2. To test this possibility, we bacterially expressed and purified the 37aa CCCD region (residues 193–229)—as predicted by the COILS server (*Lupas et al., 1991*)—as a His-tagged diLipoyl peptide (*Figure 2A,B*) (*Cottee et al., 2013*). A SEC-MALS analysis revealed that the purified protein, either with or without the Lipoyl tags, formed a tetramer at a wide range of concentrations (36–900 µM) (*Figure 2B*; Figure 4A). The CCCD tetramer was very stable and we could not find in-solution conditions under which it was dissociated, so we could not calculate a Kd. Even when examined using the usually

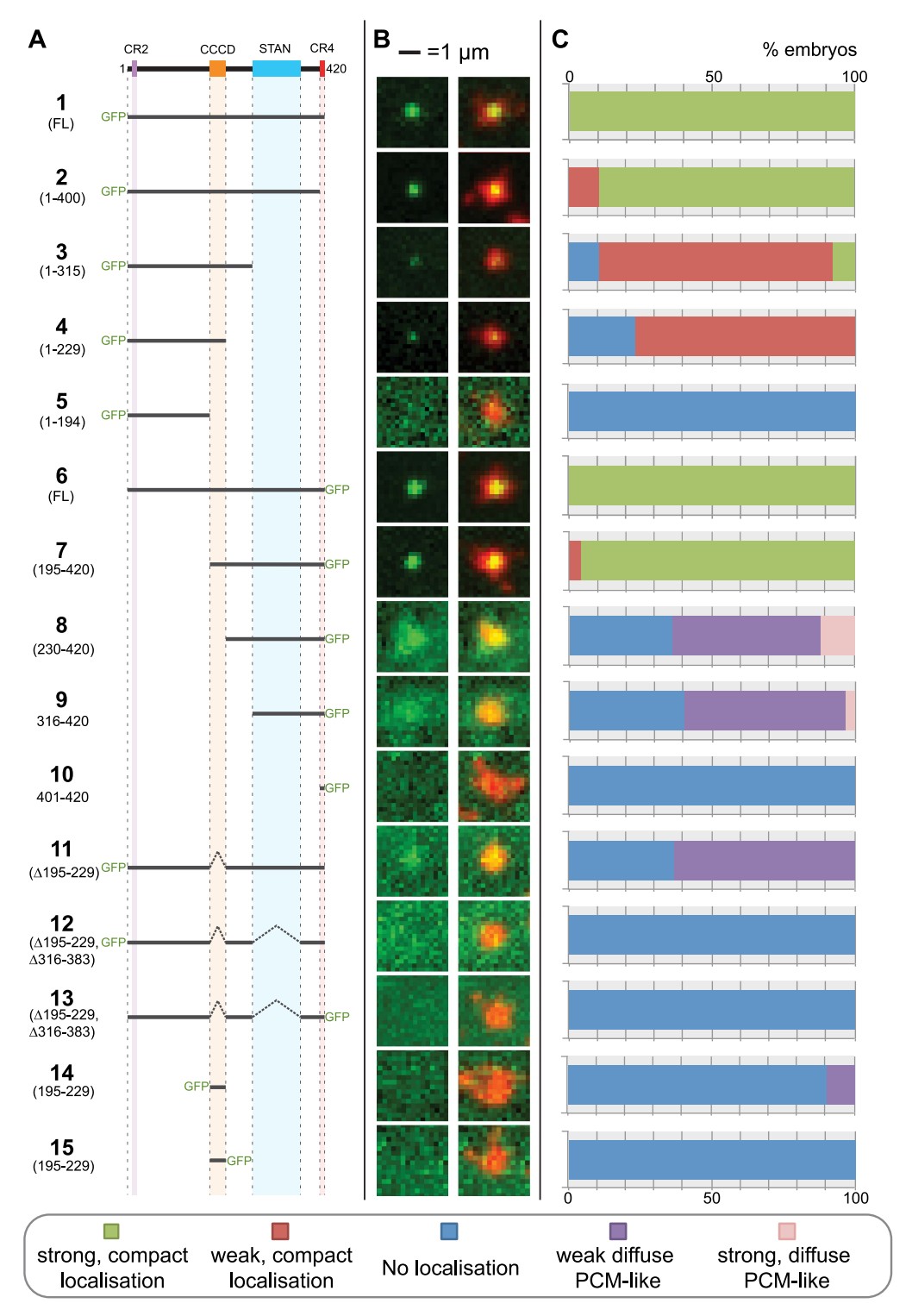

**Figure 1**. A structure/function analysis of *Drosophila* Ana2. (**A**) A schematic representation of *Drosophila* Ana2 highlighting the conserved domains and illustrating the GFP constructs analysed in this study. In vitro transcribed mRNA encoding each of these constructs was injected into *Drosophila* embryos expressing the PCM marker, RFP-Cnn; the distribution of each fusion protein was analysed in living embryos. (**B**) Micrographs show examples of typical centrosomes in embryos injected with the Ana2 constructs shown in (**A**). The localisation of the GFP-fusion protein (green) is shown on its own (left panel) and merged with RFP-Cnn (right panel). (**C**) Bars quantify the
*Figure 1. continued on next page*

*Figure 1. Continued*
localisation behaviour of the various GFP-fusions. Images of 30–80 embryos were analysed for each construct. Images of each embryo were collected and then manually sorted into various categories based on the centrosomal localisation of the GFP-fusion construct (see colour table at bottom of figure). All sorting was performed blind.

denaturing technique, Electrospray-Ionisation Mass Spectrometry, the tetramer did not fully disassemble (*Figure 2—figure supplement 1*). We also expressed and purified the 42aa predicted CCCD (residues 717–758) from the human STIL protein as a His-tagged diLipoyl peptide. This also formed a tetramer, although this was less stable than the fly CCCD tetramer and only formed at higher protein concentrations (*Figure 2C*).

## Crystal structure of the Ana2 CCCD

The purified Ana2 CCCD protein readily formed protein crystals that diffracted extremely well, enabling us to refine a structure to 0.80 Å resolution (*Figure 3*, *Figure 3—figure supplement 1*, *Table 1*). The structure demonstrated that the Ana2 CCCD forms a parallel, symmetrical 4-helix bundle, with a left-handed supercoil (*Figure 3A*). This structure appears to be unusual as we could find only one other natural soluble protein in the PDB that homo-tetramerises through a parallel four-helical bundle (NSP4, a tetrameric enterotoxin secreted by rotaviruses). Analysis using the PISA server (*Krissinel and Henrick, 2007*) showed that residues located at the *g*, *a*, *d* and *e* positions of the helical heptad repeat were all buried at the tetramer interface (*Figure 3A*, *yellow* residues). The tetramer is stabilised by at least three mechanisms: first, the knob-into-holes and van der Waals packing of hydrophobic residues (*Figure 3B,C*); second, the packing of internally facing polar residues (*Figure 3D*); third, a cross-chain salt bridge formed between R208 and E210 (*Figure 3E*).

## Mutations that perturb Ana2 tetramerisation in vitro perturb centriole duplication in vivo

To test the potential importance of tetramerisation of the CCCD in vivo, we created point mutations within the CCCD that our structural studies suggested would disrupt the ability of the CCCD to tetramerise. We replaced all ten residues at the *d* and *g* positions of the CCCD with either Ala (CCCD-A), Ser (CCCD-S) or Asp (CCCD-D) (*Figure 3A*, residues circled in *red*). A SEC-MALS analysis revealed that all of these mutant CCCD proteins behaved as monomers rather than tetramers in vitro (*Figure 4A*). We then made equivalent CCCD mutations within the context of the full length Ana2 protein and tested their localisation in our embryo RNA injection assay. All three mutant proteins were undetectable at centrioles but still localised diffusely to the PCM (*Figure 4B–D*), indicating that the mutant proteins are not simply misfolded or degraded, as the STAN domain can still target them to the PCM.

We next generated stable *Drosophila* transgenic lines that express full length Ana2-GFP containing the CCCD-A mutations under the control of the ubiquitin promoter (Ana2-CCA-GFP). This promoter consistently results in the strong overexpression of both WT Ana2-GFP and mutant Ana2-CCA-GFP relative to the endogenous protein (*Figure 5A*). While Ana2-GFP strongly rescued the centriole duplication defect seen in *ana2* mutants, Ana2-CCA-GFP rescued much more weakly, although at least one centrosome-like structure (CLS) was detectable in ~35% of cells expressing one copy of the transgene (*Figure 5B,F′,F′′*). We do not know if these structures contain *bona fide* centrioles, but they stained for multiple centriole/centrosome markers and were almost invariably located at the spindle poles in mitotic cells, demonstrating that they retain at least some centriole and centrosome function (*Figure 5F–F′′*; data not shown); we therefore refer to these structures as CLSs. Interestingly, doubling the dosage of the Ana2-CCA-GFP, which already appeared to be overexpressed even with one gene dose (*Figure 5A*), increased the efficiency of rescue, and nearly 90% of cells now contained at least one CLS (*Figure 5B*). Several of these flies were clearly less uncoordinated than the *ana2* mutant flies (data not shown), strongly suggesting that flies rescued by a double dose of Ana2-CCA-GFP can form at least some functional cilia, again arguing that the CLSs retain some centriole activity. Taken together, these observations demonstrate that the ability of Ana2 to tetramerise is important for Ana2 function and for centriole assembly, but that Ana2-CCA retains some residual ability to promote the assembly of CLSs in vivo.

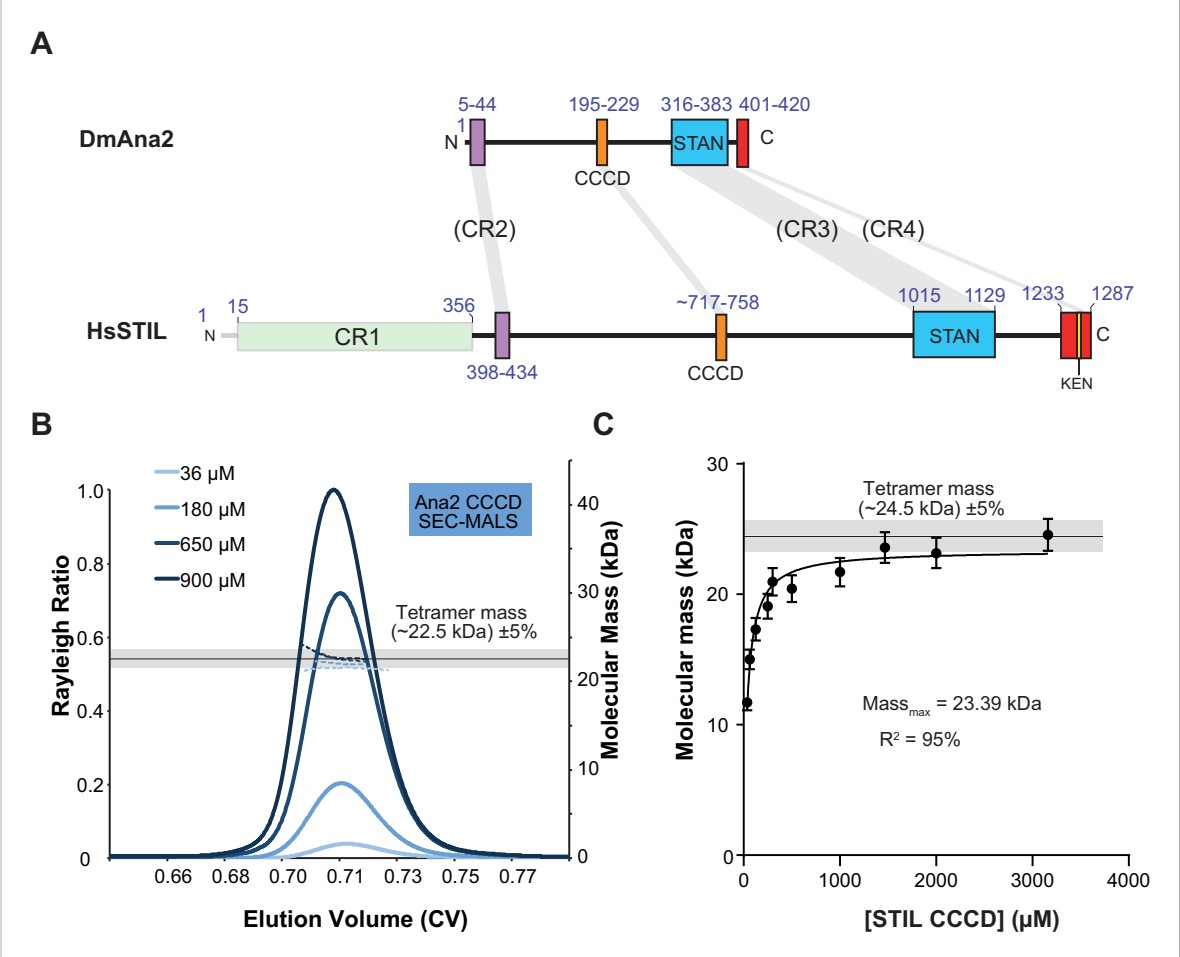

**Figure 2**. The Ana2 and STIL central coiled-coil domain (CCCD) regions form tetramers in solution. (**A**) A schematic representation of *D. melanogaster* Ana2 and human STIL highlighting the conserved domains. Note that vertebrate STIL proteins contain a conserved region 1 (CR1) of unknown function that is not present in Ana2. (**B**) A SEC-MALS analysis of the *Drosophila* Ana2 CCCD (aa193–229) was performed. Injected protein concentrations are indicated by different shades of blue—solid lines show the relative Rayleigh ratio, dashed lines the observed mass. The black horizontal line indicates the theoretical mass for an Ana2 CCCD tetramer, the grey bar indicates a ±5% tolerance. (**C**) An analysis of the observed mass of human STIL CCCD (717–758) at various injected protein concentrations obtained from SEC MALS experiments. Error bars represent an estimated ±5% error in the MALS mass measurement, as each data point represents a single injection and mass measurement. The black line and grey bar represent the theoretical tetramer mass ±5% tolerance. The data were fitted to a hyperbolic function in Graphpad Prism 6.01, including a 5% SEM for each mass value, with no extrapolation. This fitting estimated that the STIL CCCD was tending towards a mass of 23.4 kDa (theoretical tetramer mass = 24.5 kDa) with an $R^2$ value of 95%.

The following figure supplement is available for figure 2:

**Figure supplement 1**. Electrospray-ionisation mass spectrum of the Ana2 CCCD.

## *Drosophila* Sas-6 can homo-oligomerise to form a canonical cartwheel structure

It has previously been shown that Sas-6 proteins also need to homo-oligomerise to function in centriole duplication (*Kitagawa et al., 2011*; *van Breugel et al., 2011*), so we wanted to explore the relative importance of Sas-6 and Ana2 oligomerisation for centriole duplication. In all species examined to date Sas-6 forms dimers through an extended C-terminal coiled-coil region (C–C) (*Kitagawa et al., 2011*; *van Breugel et al., 2011*; *Qiao et al., 2012*). In *Danio rerio*, *Chlamydomonas* and *Leishmania* these dimers can further homo-oligomerise through an N-terminal headgroup interaction (N–N) to form a flat ninefold symmetric ring from which the C–C domains emanate—thus forming the central hub and spokes of the cartwheel (*Figure 6I*). In *Caenorhabditis elegans*, however,

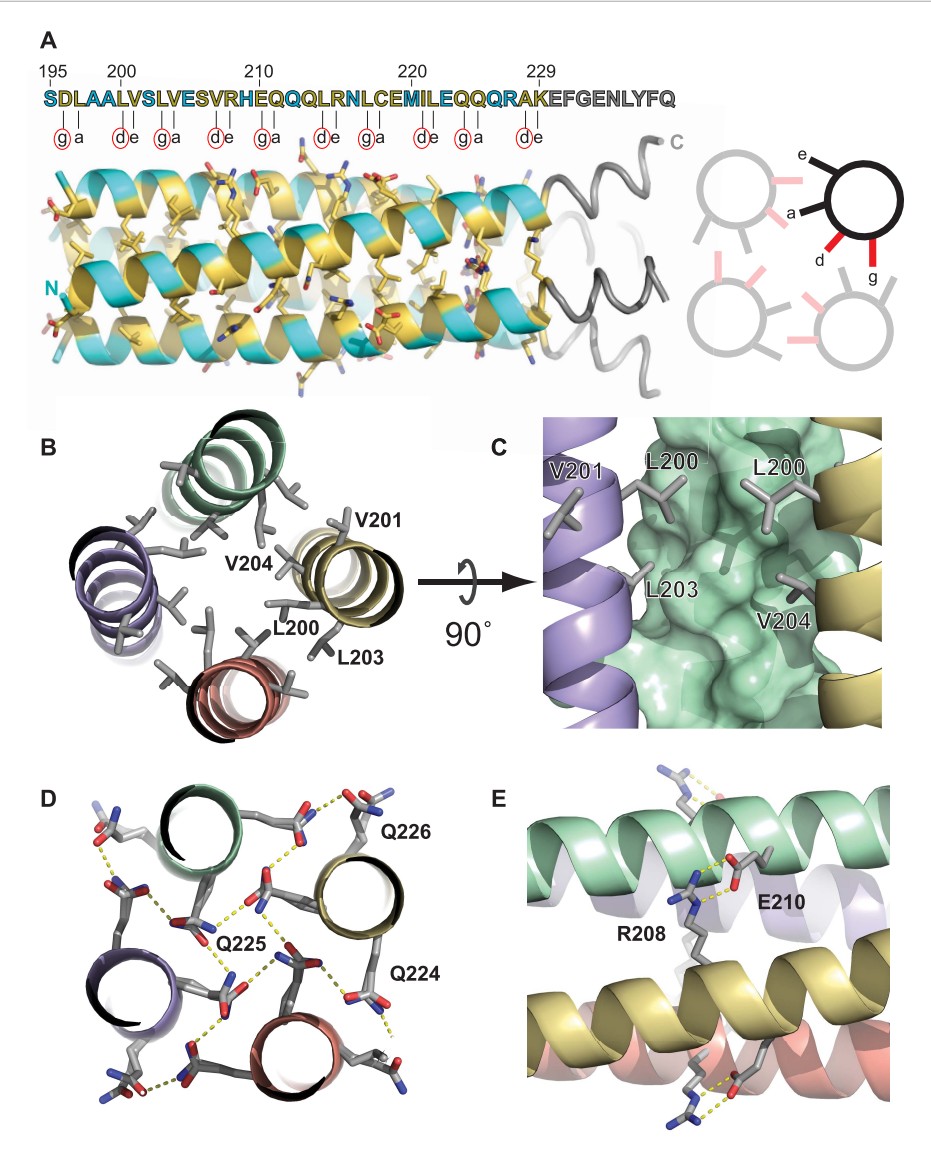

**Figure 3**. The Ana2 CCCD forms a parallel four helical tetramer. (**A**) *Left*, the structure of the Ana2 CCCD tetramer generated around the crystallographic fourfold symmetry axis. The primary amino acid sequence is shown above the structure; residues in the *g*, *a*, *d* and *e* positions of the helical heptad repeat are indicated below the sequence. All these residues were ≥30% buried (according to PISA server analysis) and are coloured in *yellow*, with side-chains in stick format—other residues are coloured in *cyan* (side-chains not shown). The TEV cleavage remnant is shown in *grey*. *Right*, schematic transverse view of the tetramer indicating how the *g*, *a*, *d* and *e* residues of the heptad repeat are buried at the tetramer interface. Note that the *g* and *d* residues (coloured *red*, and highlighted with a *red circle* underneath the primary amino acid sequence) form one side of this interface; these 10 residues were mutated to generate forms of the protein that could no longer form tetramers (see main text). (**B**–**E**) Schematics illustrate the molecular determinants of tetramerisation, with interfacing residues shown as grey sticks. (**B**) A hydrophobic cluster of interface residues. The labelled residues sit at the *g*, *a*, *d* and *e* positions of the heptad repeat, and pack closely forming a hydrophobic environment. (**C**) A side on view of the same cluster, with one chain shown as a surface. (**D**) A transverse N-C view of a QQQ triad which adopts positions *g*, *a* and *b* of the heptad. These polar side-chains form an inward facing hydrogen-bond network. (**E**) A side-on view showing a salt bridge between adjacent chains of the tetramer.

The following figure supplement is available for figure 3:

**Figure supplement 1**. Representative electron density for the Ana2 CCCD crystal structure at 0.8 Å resolution.

**Table 1**. Ana2 CCCD dataset and refinement statistics

Dataset statistics

| | |
|---|---|
| Beamline | Diamond I03 |
| Wavelength (Å) | 0.7293 |
| Spacegroup | I4 |
| Unit cell dimensions (Å/°) | 33.27, 33.27, 74.49/90.00, 90.00, 90.00 |
| Resolution (Å) (overall/inner/outer) | 30.36–0.80/30.36–3.58/0.82–0.80 |
| Completeness (overall/inner/outer) | 97.8/99.9/79.2 |
| $R_{merge}$ (overall/inner/outer) | 0.038/0.034/0.547 |
| $R_{pim}$ (overall/inner/outer) | 0.013/0.011/0.312 |
| CC (1/2) (overall/inner/outer) | 1.00/0.999/0.782 |
| I/σI (overall/inner/outer) | 21.5.71.3/2.2 |
| Mulitiplicity (overall/inner/outer) | 5.8/11.2/3.3 |
| Refinement statistics (parentheses = highest res shell) | |
| Resolution range (Å) | 30.36–0.80 (0.82.0.80) |
| $R_{work}$/$R_{free}$/% test set size | 10.6/11.6/5.06% (21.1/20.3/4.84%) |
| Number of reflections working set/test set | 39,348 (2338)/2099 (119) |
| Number of atoms (non-H) | 499 |
| Waters | 53 |
| Rmsd from ideal values: bond length (Å)/angles (°) | 0.025/2.230 |
| Average B factor (Å²) | 10.70 |
| Ramachandran outliers | 0% |
| Ramachandran favoured | 100% |
| MolProbity score (N number, percentile) | 1.22 (222, 88%) |

Ramachandran and Molprobity scores were calculated using MolProbity (*Chen et al., 2010*).

the SAS-6 headgroup-CC orientation is altered (*Figure 6H*), and SAS-6 dimers appear to oligomerise into a spiral, rather than a flat-ring (*Hilbert et al., 2013*), potentially explaining why a classical cartwheel with nine spokes has not been visualised by EM in *C. elegans* centrioles (*Pelletier et al., 2006*). In *Drosophila* centrioles, EM images reveal a clear central cartwheel hub from which emanating spokes are often visible—but it is difficult to visualise more than a few spoke structures at any one time (e.g., *Callaini et al., 1997*; *Roque et al., 2012*; Helio Roque, *personal communication*), making it unclear whether *Drosophila* Sas-6 oligomerises into a canonical ring or into a spiral. To address this issue, we attempted to examine the structure of *Drosophila* Sas-6 (*Figure 6A*).

We were unable to purify constructs containing only the N-terminal head-group, however we could purify constructs that contained the N-terminal headgroup and either 59 (Sas-6$^{1–216}$) or 84 (Sas-6$^{1–241}$) residues of the predicted C–C region. In initial attempts to purify Sas-6$^{1–241}$ the protein invariably formed large aggregates (blue trace, *Figure 6B*) that appeared to be elongated chains of protein by negative-stain EM (*Figure 6Ci–iv*). It has previously been shown that a large hydrophobic residue in the headgroup is essential for the N–N interaction in several species (*Kitagawa et al., 2011*; *van Breugel et al., 2011*), so we mutated the equivalent residue, F143, to Asp. Purified Sas-6$^{1–241}$-F143D behaved as a dimer by SEC-MALS (red trace, *Figure 6B*) and aggregates were no longer detectable by negative-stain EM (*Figure 6Cv*); we conclude that aggregate formation is dependent upon the N–N interaction, and the F143D mutation perturbs this interaction in vitro.

To investigate how *Drosophila* Sas-6 might oligomerise into a cartwheel we solved the crystal structure of Sas-6$^{1–216}$-F143D to 2.9 Å (*Figure 6D*, *Table 2*). The asymmetric unit contained a dimer of Sas-6, associated via the coiled-coil interface. To assess whether this Sas-6 N-CC dimer could be built into a canonical flat ring structure, we compared it to other Sas-6 orthologues for which structures are

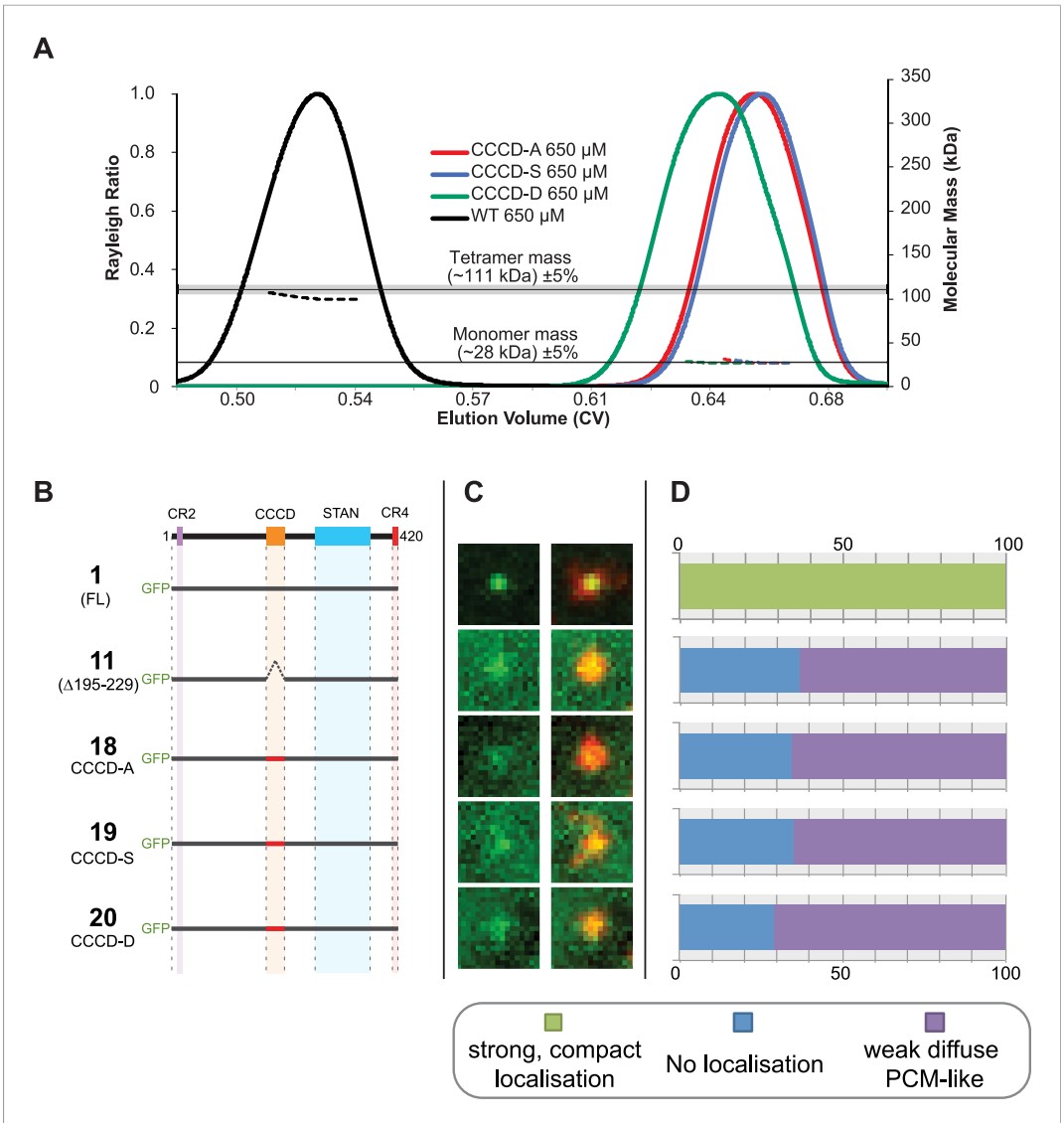

**Figure 4**. Mutations of the CCCD that perturb tetramer formation in vitro perturb the localisation of Ana2 to centrioles in vivo. (**A**) A SEC-MALS analysis of wild type (WT) and mutant forms of the CCCD where the 10 *d* and *g* residues important for tetramer formation (circled in *red*, *Figure 3A*) have been mutated either to Ala (CCCD-A), Ser (CCCD-S) or Asp (CCCD-D). Horizontal black lines illustrate the theoretical molecular mass of a tetramer and monomer, grey shading represents ±5% tolerance. Note that, in contrast to the SEC-MALS analysis presented in *Figure 1A*, the diLipoyl domains of the fusion proteins have not been removed in this experiment, so the masses of the monomer and tetramer are higher. (**B**) A schematic representation of the GFP-Ana2 fusions that contain mutations of the CCCD (constructs #1 and #11 are the same constructs shown in *Figure 1A*). In vitro transcribed mRNA encoding each of these constructs was injected into Drosophila embryos expressing the PCM marker, RFP-Cnn; the distribution of each fusion protein was analysed in living embryos. (**C**) Micrographs show examples of typical centrosomes in embryos injected with the Ana2 constructs shown in (**A**). The localisation of the GFP-fusion protein (green) is shown on its own (left panel) and merged with RFP-Cnn (right panel). (**D**) Bars quantify the localisation behaviour of the various GFP-fusions. Images of 34–40 embryos were analysed for each construct. Images of each embryo were collected and then manually sorted into various categories based on the centrosomal localisation of the GFP-fusion construct (see colour table at bottom of figure). All sorting was performed blind. The data shown here for constructs #1 and #11 is the same as that presented in *Figure 1C*.

available. The *Dm*Sas-6 N-CC dimer could be superimposed with Sas-6 N-CC dimers from *D. rerio*, *Chlamydomonas* and *Leishmania* Sas-6 (average pairwise RMSD 1.87 ± 0.31 Å over 617 ± 47 backbone atom pairs) (*Figure 6E–G*). However it could not be superimposed onto *C. elegans* SAS-6,

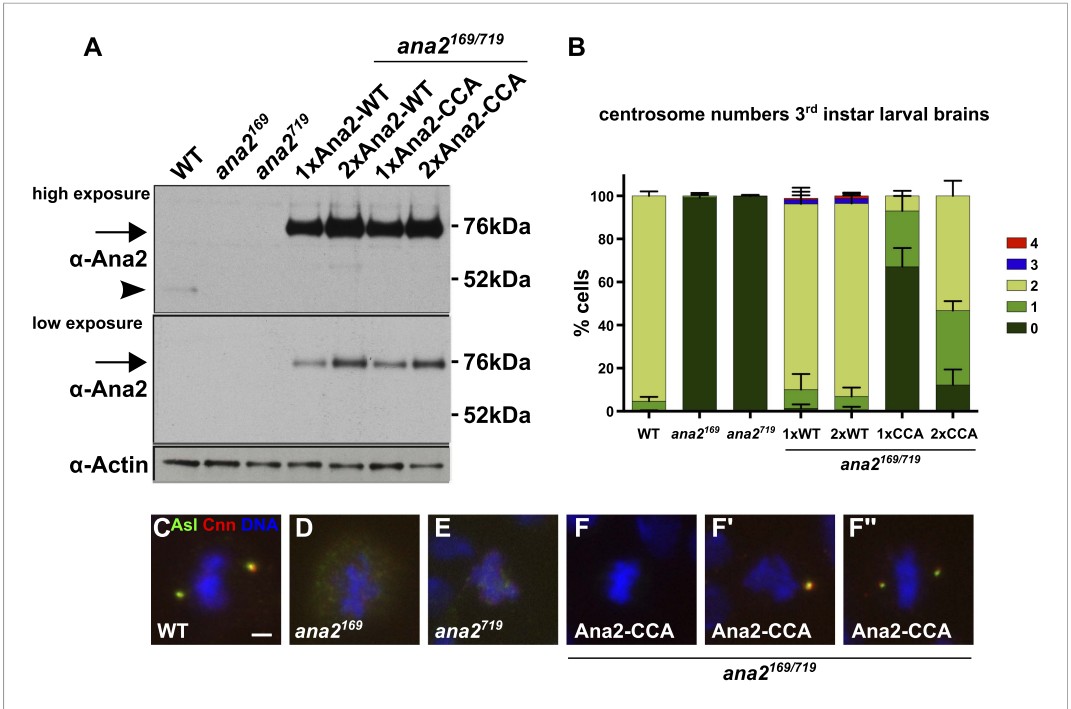

**Figure 5**. A Mutant form of Ana2 that cannot tetramerise efficiently in vitro cannot support efficient centriole duplication in vivo. (**A**) Two exposures of a western blot illustrating the relative expression levels of endogenous Ana2 (arrowhead) and either WT Ana2-GFP or Ana2-CCA-GFP (arrow) (expressed from one (1×) or two (2×) copies of the transgene) in third instar larval brains in either a WT or *ana2* mutant background. Actin is shown as a loading control. (**B**) The bar chart shows the number of centrosomes or centrosome-like structures (CLSs) observed in mitotic third instar larval brain cells (scored by the presence of both the centriole marker Asl and the centrosome marker Cnn) in WT, *ana2* mutant and *ana2* mutants expressing one (1×) or two (2×) copies of either WT Ana2-GFP or Ana2-CCA-GFP, as indicated. A total of at least 300 mitotic cells from at least five different brains were scored for each genotype; error bars represent the SD. (**C–F''**) Micrographs show the distribution of Asl (green) and Cnn (red) in representative mitotic third instar larval brain cells of the indicated genotypes. DNA is in blue. The images in **F–F''** show cells rescued with the Ana2-CCA-GFP construct that have either no centrosomes (**F**) or one (**F'**) or two (**F''**) CLSs. Scale bar in C: 2 μm.

which has an alternative head-group-spoke conformation (*Figure 6H*). Furthermore, we found that the *Dm*Sas-6 N-CC dimer could be modelled into a flat ninefold ring (*Figure 6I*), similar to that observed in crystallo for *Leishmania* Sas-6 (*van Breugel et al., 2014*). The structure of *Dm*Sas-6 is therefore highly similar to Sas-6 orthologues from organisms with canonical cartwheels, suggesting that it also forms such a structure.

## Mutations that perturb Sas-6 oligomerisation in vitro perturb centriole duplication in vivo

To test whether the ability of Sas-6 to form higher-order oligomers was important for Sas-6 function, as has been observed in several other systems (*Kitagawa et al., 2011*; *van Breugel et al., 2011*), we generated stable transgenic lines expressing either WT GFP-Sas-6 or GFP-Sas-6-F143D under the control of the ubiquitin promoter. This promoter consistently resulted in the overexpression of both WT GFP-Sas-6 and GFP-Sas-6-F143D compared to the endogenous protein (*Figure 7A*). While WT GFP-Sas-6 strongly rescued the centriole duplication defect seen in *Sas-6* mutants, GFP-Sas-6-F143D rescued much more weakly, although, at least one CLS was detectable in ~60% of cells expressing one copy of the transgene (*Figure 7B–F''*). As was the case with the rescue of the *ana2* mutation by Ana2-CCA-GFP, these structures stained for multiple centriole/centrosome markers and were usually located at the spindle poles in mitotic cells, demonstrating that they retain at least some centriole and centrosome function (*Figure 7F–F''*; data not shown). From our qualitative analysis, however, the

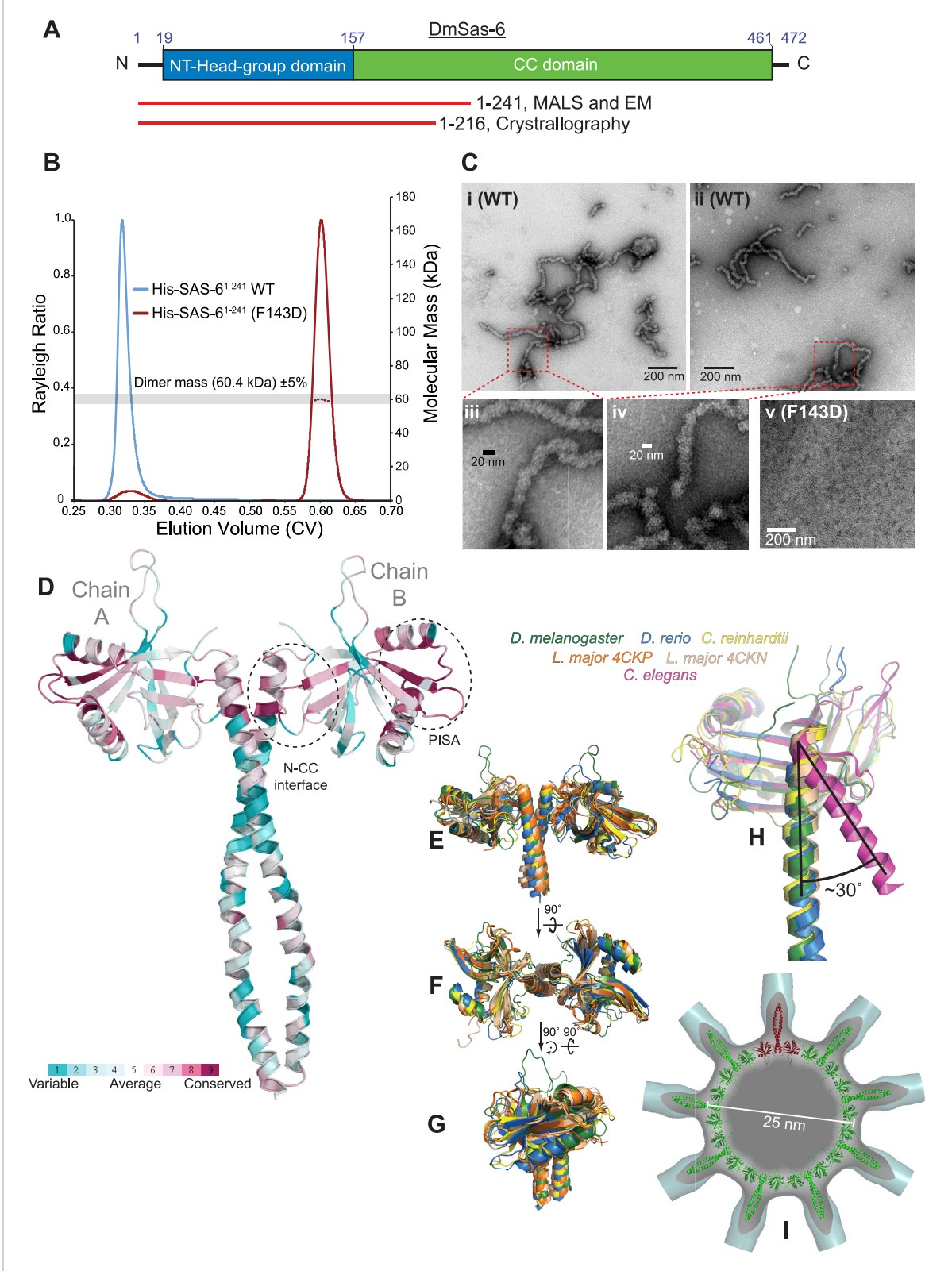

**Figure 6**. A biochemical and structural analysis of *Drosophila* Sas-6. (**A**) A schematic representation of *Drosophila* Sas-6 highlighting the position of the N-terminal head domain (blue) and C-terminal coiled-coil (CC) domain (green). Red lines below represent the constructs used in SEC-MALS and EM studies (top) and in X-Ray Crystallography studies (bottom). (**B**) A SEC-MALS analysis of WT (blue trace) and F143D mutant (red trace) Sas-6$_{1-241}$ proteins, injected

*Figure 6. continued on next page*

*Figure 6. Continued*

at 33 µM. The horizontal black line and grey bar represent the theoretical dimer mass ±5% tolerance. The WT protein could not be analysed by MALS as it eluted in the void volume and appeared to form a range of higher-order oligomers. (**C**) Negative-stain EM analysis of purified WT ([i]–[iv]) Sas-6[1–241] protein, showing the chain-like structures formed ([iii] and [iv] show magnified views of the red boxed areas in [i] and [ii]); these structures are not detectable in preparations of the mutant Sas-6-F143D[1–241] protein ([v]). (**D**) The structure of the Sas-6 dimer, coloured according to Consurf conservation scores (*Glaser et al., 2003*) from *cyan* (variable) to *burgundy* (conserved). The conserved PISA domain and the N-CC interface regions are highlighted with dashed circles. (**E**–**G**) Superimposed structures from *D. melanogaster, D. rerio, Chlamydomonas* and *Leishmania* (as indicated) of the Sas-6 N-terminal head-group with a short stretch of the coiled-coil domain. (**H**) Superimposed structures of the N-CC interface in *D. melanogaster, D. rerio, Chlamydomonas* and *C. elegans.* Note how the interface is rotated by ∼30° in *C. elegans* (*purple*) compared to the other structures. (**I**) The *DmSas-6* structure modelled into a ninefold symmetric flat ring (*green*, single dimer shown in *red*), similar to that observed in crystallo for *LmSAS-6*. This ring structure was docked into the EM density of the *Triconympha* cartwheel structure (*Guichard et al., 2013*) (*cyan* surface, cut away to reveal the *DmSAS-6* ring).

CLSs formed when *Sas-6* mutants were rescued by GFP-Sas-6-F143D often appeared smaller and more fragmented than those observed when *ana2* mutants were rescued by Ana2-CCA-GFP, suggesting that the CLSs formed in the presence of GFP-Sas-6-F143D may be less well organised than those formed in the presence of Ana2-CCA-GFP. Moreover, as described below, females carrying even one copy of this transgene invariably laid embryos that arrested early in development, so we could not generate flies carrying two copies of the transgene to test if the rescuing activity of the transgene increased with gene dosage—as we observed for Ana2-CCA-GFP (*Figure 5B*). Nevertheless, these data demonstrate that the ability of Sas-6 to form higher order oligomers is important for

**Table 2**. Sas-6[1–216] (F143D) dataset and refinement statistics

| Dataset statistics | |
|---|---|
| Beamline | ESRF ID23-2 |
| Wavelength (Å) | 0.8726 |
| Spacegroup | P2 |
| Unit cell dimensions (Å/°) | 47.13, 64.74, 123.73/90.00, 98.91, 90.00 |
| Resolution (Å) (overall/inner/outer) | 41.43–2.92/41.43–13.06/3.00–2.92 |
| Completeness (overall/inner/outer) | 97.4/90.4/96.4 |
| $R_{merge}$ (overall/inner/outer) | 0.128/0.035/0.668 |
| $R_{pim}$ (overall/inner/outer) | 0.060/0.017/0.318 |
| CC (1/2) (overall/inner/outer) | 0.994/0.988/0.799 |
| I/σI (overall/inner/outer) | 12.1/48.4/2.6 |
| Mulitiplicity (overall/inner/outer) | 5.2/4.8/5.1 |
| Refinement statistics (parentheses = highest res shell) | |
| Resolution range (Å) | 41.43–2.92 (3.10–2.92) |
| $R_{work}/R_{free}$/% test set size | 18.3/21.5/5.00% (26.2/34.5/5.90%) |
| Number of reflections working set/test set | 14,976 (2456)/788 (154) |
| Number of atoms (non-H) | 3405 |
| Waters | 33 |
| Rmsd from ideal values: bond length (Å)/angles (°) | 0.007/1.055 |
| Average B factor (Å²) | 76.30 |
| Ramachandran outliers | 0% |
| Ramachandran favoured | 94.9% |
| Molprobity score (N number, percentile) | 1.54 (3648, 100%) |

Ramachandran and Molprobity scores were calculated using MolProbity (*Chen et al., 2010*).

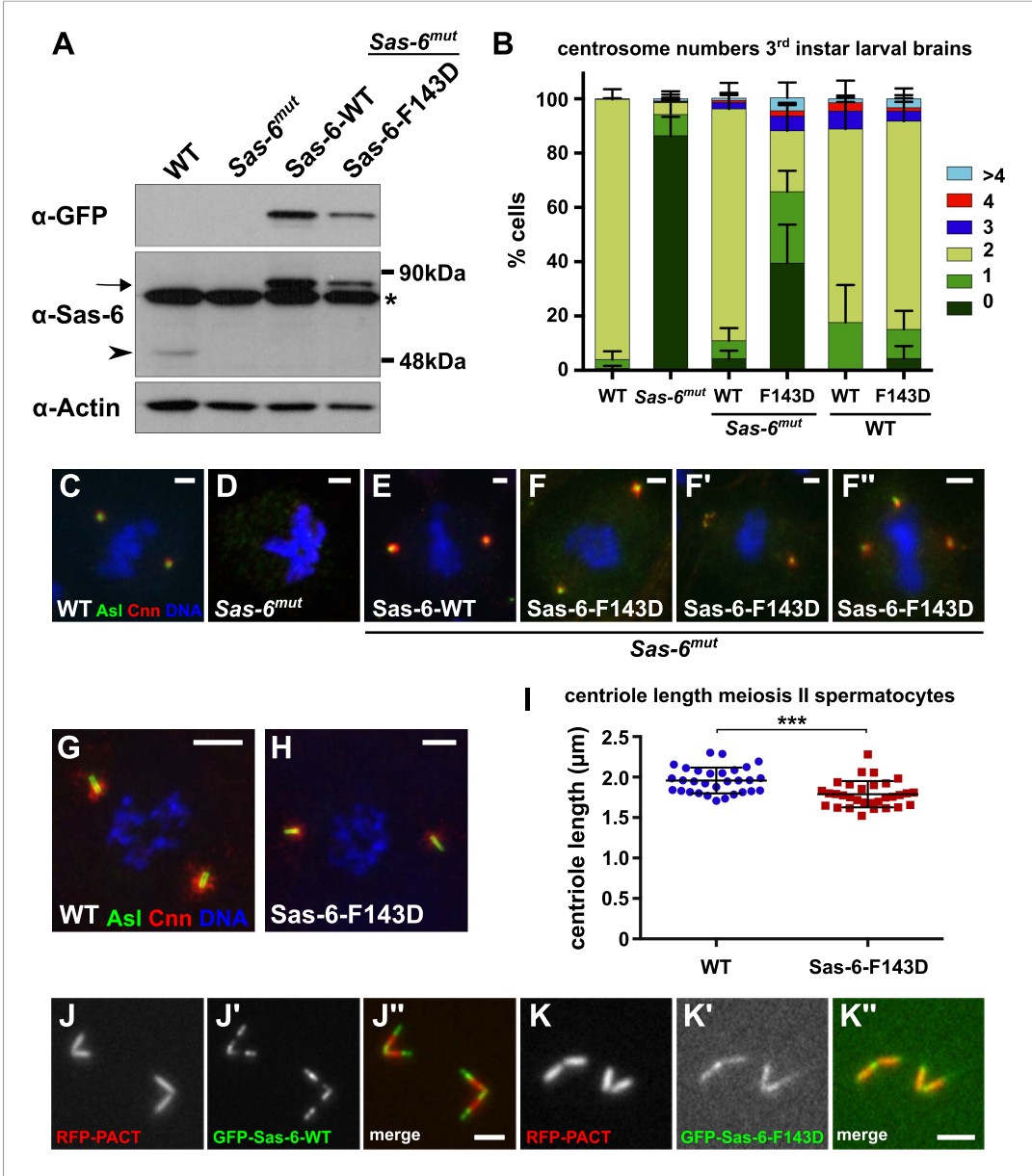

**Figure 7**. A Mutant form of Sas-6 that cannot oligomerise efficiently in vitro cannot support efficient centriole duplication in vivo. (**A**) A western blot illustrating the relative expression levels of endogenous Sas-6 (arrowhead) and either WT GFP-Sas-6 or GFP-Sas-6-F143D (arrow) in third instar larval brains in either a WT or *Sas-6* mutant background. Actin is shown as a loading control, and an (*) marks a non-specific band. (**B**) The bar chart shows the number of centrosomes or CLSs observed in mitotic third instar larval brain cells in WT or *Sas-6* mutants (first and second bars); in *Sas-6* mutants expressing either WT GFP-Sas-6 or GFP-Sas-6-F143D (third and fourth bars); or WT brains expressing either WT GFP-Sas-6 or GFP-Sas-6-F143D (fifth and sixth bars). At least 600 mitotic cells from at least five different brains were scored for each genotype; error bars represent the SD. (**C–F''**) Micrographs show the distribution of Asl (green) and Cnn (red) in representative mitotic third instar larval brain cells of the indicated genotypes. DNA is in blue. The images in **F**, **F'** and **F''** show *Sas-6* mutant cells rescued by Sas-6-F143D showing examples of the CLSs. (**G, H**) Micrographs show the distribution of Asl (green) and Cnn (red) in either a WT primary spermatocyte (**G**) or a WT primary spermatocyte overexpressing GFP-Sas-6-F143D (**H**). (**I**) Graph shows the quantification of centriole length (as measured by Asl staining) in WT primary spermatocytes (blue circles) or WT primary spermatocytes overexpressing GFP-Sas-6-F143D (red boxes); at least 1500 centrioles from at least 30 different testes were scored for each genotype, and each circle or box represents the mean from an individual testes. Statistical significance was assessed using an unpaired two-tailed t-test: (***) indicates p-value < 0.001. (**J–K**) Micrographs show the distribution of the centriole marker RFP-PACT (red) and either WT GFP-Sas-6 (green) (**J–J''**) or GFP-Sas-6-F143D (**K–K''**) in WT primary spermatocytes. Scale bars: 2 µm in **C–F''** and **J–K''** and 5 µm in **H–I**.

Sas-6 function and for centriole assembly, but that GFP-Sas-6-F143D retains some residual ability to promote the assembly of CLSs in vivo (*Figure 6B,C*).

## GFP-Sas-6-F143D exhibits a dominant-negative effect on centriole duplication in early embryos, but not in several other cell types

In light of the proposed mechanism of Sas-6-supported cartwheel assembly, the overexpression of mutant forms of the protein that cannot form higher order oligomers through the N–N interaction might be expected to act as dominant-negatives, capable of 'poisoning' cartwheel assembly by forming hetero-dimers with the WT protein that can incorporate into the cartwheel through the WT headgroup, but which cannot then interact with another headgroup—thus blocking further cartwheel assembly (*Figure 8A*). Surprisingly, however, although GFP-Sas-6-F143D was overexpressed in all tissues we examined (embryos, brains and testes), it had very little, if any, negative effect on centriole duplication in WT brain cells (*Figure 7B*) or spermatocytes (*Figure 7G,H*), although the centrioles were ~10% shorter in spermatocytes overexpressing GFP-Sas-6-F143D (*Figure 7I*). Interestingly, small amounts of GFP-Sas-6-F143D could be detected in spermatocyte centrioles, but this was more diffusely localised throughout the centriole length when compared to the WT GFP-Sas-6, which was strongly concentrated at the proximal and distal ends of the centrioles (as reported previously) (*Peel et al., 2007*) (*Figure 7J,K*). Moreover, WT flies overexpressing GFP-Sas-6-F143D were not noticeably uncoordinated, demonstrating that they can form functional cilia.

In embryos, however, GFP-Sas-6-F143D had a strong dominant-negative affect, and females expressing this transgene laid embryos that invariably arrested early in development after they had gone through only a few rounds of nuclear division (*Figure 8B,C*). The MTs in these embryos appeared to be organised by centrioles that had incorporated only very small amounts of GFP-Sas-6-F143D (*Figure 8D,E*), and which often appeared small and fragmented (*Figure 8F–H*). These observations have important implications for the mechanism of Sas-6-mediated cartwheel assembly as they suggest that GFP-Sas-6-F143D can effectively poison cartwheel assembly in rapidly dividing syncytial embryos that have to assemble centrioles very quickly (and centrioles are essential for early embryo development in flies [*Stevens et al., 2007*; *Varmark et al., 2007*]), but not in brain cells or spermatocytes that have a slower cell cycle and so can presumably assemble their centrioles over a longer time-frame (see 'Discussion').

## Discussion

It is now widely accepted that the structure of the centriole cartwheel is formed around a core of 9 Sas-6 dimers that homo-oligomerise to form a ring structure (*Cottee et al., 2011*). Sas-6 molecules can form such ninefold symmetric rings in the absence of any other proteins in vitro, and mutations that perturb the ability of Sas-6 to homo-oligomerise in vitro strongly perturb centriole assembly in vivo (*Kitagawa et al., 2011*; *van Breugel et al., 2011*, *2014*). We previously showed, however, that overexpressed Sas-6 can only form cartwheel-like structures in fly spermatocytes when Ana2 is also overexpressed (*Stevens et al., 2010b*). Here we show that Ana2/STIL proteins also homo-oligomerise and that mutations that perturb the homo-oligomerisation of fly Ana2 in vitro also strongly perturb centriole assembly in vivo. Thus, Sas-6 homo-oligomerisation alone appears unable to drive efficient cartwheel assembly in vivo if Ana2 is unable to homo-oligomerise.

Our initial structure/function analysis of Ana2 revealed that the CCCD is important for the recruitment of Ana2 to centrioles. This is consistent with the recent discovery that the CCCD in human STIL interacts with Plk4 and is important for STIL recruitment to centrioles (*Ohta et al., 2014*; *Kratz et al., 2015*). In both flies and vertebrates, Sak/Plk4 can also phosphorylate the STAN domain of Ana2/STIL, promoting its interaction with Sas-6, and allowing it to recruit Sas-6 to the newly forming centriole (*Dzhindzhev et al., 2014*; *Ohta et al., 2014*; *Kratz et al., 2015*). Interestingly, we found that although the STAN domain could not localise Ana2 to centrioles in fly embryos in the absence of the CCCD, the centriolar localisation of Ana2 was much weaker in the absence of the STAN domain, suggesting that an interaction with Sas-6 is required for robust Ana2 localisation in flies. This result is in agreement with the finding that fragments of STIL containing both the CCCD and the STAN domain interact most strongly with Plk4 (*Kratz et al., 2015*), but contrasts with reports in human cells where deleting the STAN domain did not affect STIL localisation to centrioles (*Ohta et al., 2014*) and in fly cultured cells, where the depletion of Sas-6 by RNAi did not detectably perturb the centriole

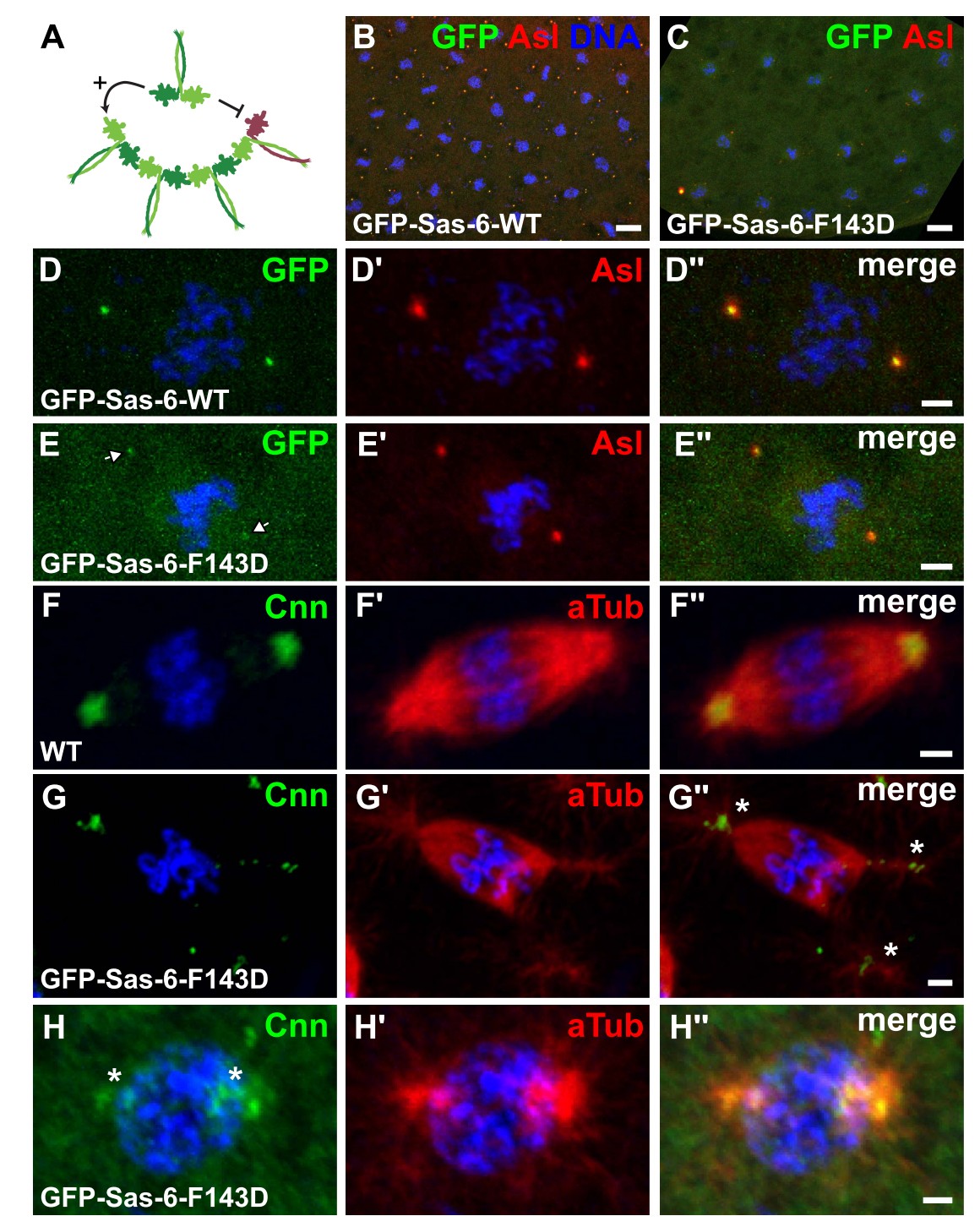

**Figure 8**. GFP-Sas-6-F143D dominantly suppresses centrosome assembly in early embryos. (**A**) A schematic illustration of how GFP-Sas-6-F143D could act as a dominant-negative in cartwheel assembly. WT Sas-6 (light and dark green) can form WT–WT homodimers or WT-mutant heterodimers with GFP-Sas-6-F143D (red). The homodimers can support cartwheel assembly while the heterodimers can incorporate into the growing cartwheel (through the WT headgroup), but cannot support further cartwheel assembly. The heterodimer must dissociate before a WT homodimer can incorporate into the cartwheel, so allowing cartwheel assembly to proceed. (**B–H''**) Micrographs show images from WT embryos expressing no transgene (**F**) or expressing either WT GFP-Sas-6 (**B**, **D**) or GFP-Sas-6-F143D (**C**, **E**, **G**, **H**) stained to reveal the distribution of GFP, Asl, Cnn or α-tubulin, as indicated. (**B**, **D**) Embryos expressing WT GFP-Sas-6 develop normally, and the fusion protein strongly localises to a bright spot in the centre of the centrosomes. (**C**, **E**) Embryos expressing GFP-Sas-6-F143D arrest during the early syncytial stages; some of these embryos are reasonably well organized and centrioles are observed at

*Figure 8. continued on next page*

Figure 8. Continued

the spindle poles, but these contain very little detectable GFP-Sas-6-F143D. (**F–G**) Most embryos are less well organized and contain abnormal microtubule (MT) arrays organized by fragmented centrosomes (**G, H**) when compared to WT (**F**). Scale bars: 10 μm in **B, C**, 2 μm in **D–D''** and **F–H''** and 3 μm in **E–E''**.

localisation of Ana2 (*Dzhindzhev et al., 2014*). These latter results suggest that the proper recruitment of Ana2/STIL to centrioles is independent of the STAN domain's interaction with Sas-6. Our findings suggest, however, that although Plk4 may initially recruit some Ana2 to centrioles without Sas-6, the subsequent incorporation of Ana2 into the new centriole is dependent upon the successful assembly of a cartwheel structure, and this cannot occur without Sas-6. In addition, although Plk4 can clearly recruit Ana2/STIL to centrioles in flies and humans, in worms ZYG-1 (the Plk4 functional homologue) can directly recruit SAS-6 (*Lettman et al., 2013*). Thus, the molecular detail of the interactions between Plk4/Sak/ZYG-1, Sas-6 and Ana2/STIL/SAS-5 involved in cartwheel assembly remain to be fully elucidated, and may vary between different cell types and species.

The Ana2 CCCD tetramerises as a symmetric, parallel four helical bundle. Such structures are relatively rare in cytoplasmic proteins: most examples in the PDB are either engineered peptides, extensions of larger domains such as tetrameric membrane-associated receptors, or occur within a single polypeptide (and so are not oligomerisation domains). Although Ana2/STIL/SAS-5 proteins are highly diverged, a CCCD is found in all family members described to date (*Figure 2A*). Our data suggests that the human STIL CCCD can also form tetramers, although it does so more weakly than the fly CCCD; we speculate that tetramerisation via the CCCD could be a common feature of Ana2/ STIL proteins. A fragment of *C. elegans* SAS-5 also behaves as a tetramer, although the relevant oligomerisation domain has not been identified (*Shimanovskaya et al., 2013*). It will be particularly interesting to test whether the *C. elegans* SAS-5 CCCD forms a tetramer, as this organism appears to form a spiral cartwheel rather than a flat-ring cartwheel (*Hilbert et al., 2013*). It was recently proposed that fly Ana2 can form tetramers through a different mechanism that is dependent on an interaction with dynein light chain (LC8) (*Slevin et al., 2014*). This interaction may be important for spindle orientation, rather than centriole duplication (*Wang et al., 2011*), and our results suggest that the observed tetramerisation was likely driven by the CCCD rather than the interaction with LC8.

We also examined the structure of *Drosophila* Sas-6 and confirmed that, similar to other Sas-6 orthologues, it associates via two self-interaction interfaces, a C-terminal coiled-coil dimerization (C–C), and an N-terminal headgroup oligomerization (N–N). The interface between the headgroup and the C–C was similar to that seen in most other species (*Kitagawa et al., 2011*; *van Breugel et al., 2011*, *2014*), and different to that observed in *C. elegans* (*Hilbert et al., 2013*) strongly suggesting that *Drosophila* Sas-6 assembles into a canonical, flat ring structure. In agreement with previous studies (*Kitagawa et al., 2011*; *van Breugel et al., 2011*), we found that mutations that perturb the ability of *Drosophila* Sas-6 to homo-oligomerise through the N–N interface in vitro (GFP-Sas-6-F143D) cannot support efficient centriole duplication in vivo, and this is also true for mutations that perturb the ability of Ana2 to tetramerise in vitro (Ana2-CCA-GFP). Most importantly, however, we note that both these mutant proteins can support the assembly of some centrosomes, or CLSs, that can recruit other centriole and centrosome proteins and that often concentrate at spindle poles. We suspect that this is because centriole assembly is normally driven by a complex set of interactions between proteins such as Plk4, Sas-6, Ana2/STIL, Sas-4 and Cep135/Bld10 so that some residual (although possibly abnormal) centriole assembly is still possible even if one of these interaction interfaces is perturbed.

We predicted that GFP-Sas-6-F143D might act as a dominant-negative in cells, forming hetero-dimers with the WT protein that can incorporate into the cartwheel through the WT subunit, but which cannot then support further cartwheel assembly (*Figure 8A*). Surprisingly, although GFP-Sas-6-F143D appears to be overexpressed in embryos, brains and testes, it only had a dominant-negative effect on centriole duplication in early embryos (although the centrioles in spermatocytes were slightly, but significantly, shorter in the presence of GFP-Sas-6-F143D). These data suggest that WT-mutant heterodimers can transiently incorporate into the cartwheel and perturb assembly but, if given enough time, the heterodimers will dissociate and eventually a cartwheel can assemble from the pool of WT–WT homodimers. This would explain why GFP-Sas-6-F143D has no dramatic dominant-negative effect in somatic cells (where centriole assembly can presumably occur relatively slowly

during an S-phase period that can last several hours) but has a dramatic dominant-negative effect in syncytial embryos (where centriole assembly must be completed during an S-phase that lasts only a few minutes). If this interpretation is correct, it implies that partially assembled cartwheel rings must be relatively stable structures that can be maintained in the partially assembled state until enough WT–WT homodimers have been incorporated to complete ring assembly.

Interestingly, overexpressed Ana2-CCA-GFP had no detectable dominant-negative effect on centriole duplication, even in rapidly dividing syncytial embryos. We speculate that this is because Ana2 is recruited to centrioles as a tetramer; thus, monomeric Ana2-CCA-GFP molecules would not efficiently compete with WT tetramers for centriole binding sites, perhaps because they lack the avidity of the tetramer. The Ana2-CCA-GFP molecules would also not 'poison' the WT molecules, as they would be unable to tetramerise with them. It is tempting to further speculate that two dimers of Sak/Plk4 (*Slevin et al., 2012*; *Park et al., 2014*; *Shimanovskaya et al., 2014*) might function as the centriole binding sites for the Ana2 tetramer.

The Sas-6 crystal structure can be modelled with high precision into the Cryo-EM tomographic map of the *Trichonympha* cartwheel structure (*Guichard et al., 2013*). Ana2/STIL proteins contain essential CR2 and STAN domains that interact with Sas-4/CPAP and Sas-6, respectively: these domains are connected to the CCCD by extended, unconserved (and probably relatively unstructured) linkers. Thus, in flies, the Ana2 CCCD parallel tetramer tethers the CR2 and STAN domains in a particular geometry and stoichiometry. In *Figure 9* we present several models of how the Ana2 tetramer might be incorporated into the *Trichonympha* cartwheel structure. An interesting feature of the cartwheel is that its basic building block is two Sas-6 rings assembled on top of each other that are held together by the convergence of their coiled-coil spokes at the outer region of the cartwheel (*Figure 9*). Every paired spoke comprises four molecules of Sas-6, so we favour the idea that the Ana2 tetramer might interact with these converged spokes, thus stabilising the basic, two ring, building block of the cartwheel (*Figure 9A,B*). If each Sas-6 ring has a strong, but not invariant, tendency to adopt a ninefold symmetric organization, having cartwheel assembly dependent on the simultaneous co-assembly of two rings, rather than just one, could dramatically increase the precision of ninefold symmetric ring assembly. Ana2 might also hold the Sas-6 molecules in an orientation that further favours the assembly of ninefold symmetric rings.

Previous studies have indicated that STIL molecules can turnover at centrioles, suggesting that they are not 'locked' into the cartwheel structure (*Vulprecht et al., 2012*). In our favoured model (*Figure 9A,B*) Ana2 tetramers would help recruit Sas-6 dimers—and presumably also Sas-4 molecules (*Cottee et al., 2013*; *Hilbert et al., 2013*)—in a stoichiometry and conformation that would favour cartwheel assembly. Once incorporated, however, Sas-6 and Sas-4 molecules could quickly become 'locked' into the assembled cartwheel/centriole structure. Thus, Ana2 could be important for ensuring proper cartwheel assembly, but may not be required to maintain the assembled cartwheel structure. Although these models are attractive, several other arrangements of the Ana2 tetramer are also plausible (*Figure 9*), and more data will be required to identify the precise inter-molecular interactions that lead to the efficient assembly of the ninefold-symmetric cartwheel structure. Nevertheless, our findings indicate that the Ana2 tetramer plays an important part in cartwheel assembly in flies.

## Note added in proof

Rogala et al. have now shown that *C. elegans* SAS-5, the functional homologue of Ana2/STIL, can also form higher-order oligomers and this is essential for SAS-5 function. SAS-5 has two major oligomerisation domains (a central coiled-coil region and an implico domain) that allow the protein to form a mix of tetramers and hexamers in vitro. This difference may reflect that SAS-6 forms a spiral, rather than a cartwheel, structure in *C. elegans* (*Rogala et al., 2015*).

## Materials and methods

### In vitro mRNA production and injection

Fragments of Ana2 were PCR amplified from cDNA and subcloned into modified pRNA destination vectors (*Conduit et al., 2014*) using the Gateway (Life Technologies, Carlsbad, CA) system. These vectors contain a T3 RNA polymerase promoter, a polyA tail and encode GFP in-frame, 5′ or 3′ of the insert. Deletion constructs were generated using a Quikchange II XL mutagenesis kit (Agilent Technologies, Santa Clara, CA). Mutation constructs (CCCD-A/CCCD-S/CCCD-D) were synthesised de novo (Genewiz, South Plainfield, NJ) using *Drosophila*-optimised codons for each substituted

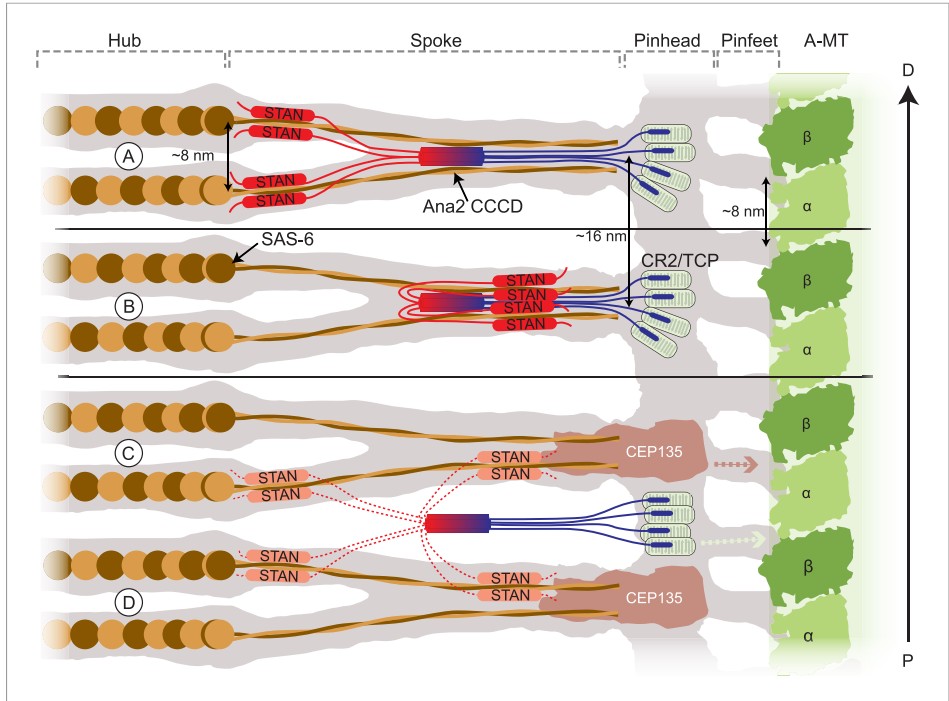

**Figure 9**. Schematic models of how Ana2 tetramers might contribute to cartwheel assembly. The grey area shows a representation of the electron density map derived from the *Triconympha* cartwheel structure (emd-2329/2330) (*Guichard et al., 2013*). A protofilament of the centriole A-MT is shown in green, and full-length Sas-6 molecules (brown/orange) have been placed into the cartwheel density. The Ana2 CCCD tetramer is coloured blue/red, extending out to the N-terminus in blue and to the C-terminus in red. The N-terminal Ana2 CR2 domain (blue) is shown bound to the Sas-4 C-terminal TCP domain (pale green) (*Cottee et al., 2013*; *Hatzopoulos et al., 2013*); Sas-4 is located towards the periphery of the cartwheel (*Mennella et al., 2012*) so we place the CR2/TCP interaction in the peripheral pinhead region. The STAN domain (red) binds Sas-6 (*Ohta et al., 2014*); as the C–C domain of Sas-6 is required for targeting to the centriole (*Keller et al., 2014*) (and so presumably for binding to Ana2). We present two alternative models where the STAN domain binds to the Sas-6 C–C region either at the N-terminal head-linker region (**A**) or at a more C-terminal region towards the end of the spoke (**B**). (**A**, **B**) In these models, the Ana2 tetramer interacts with two Sas-6 dimers, one that is incorporated into the top Sas-6 ring, and one that is incorporated into the bottom Sas-6 ring. An attractive feature of this model is that it can explain why the fundamental building block of the cartwheel seems to comprise two Sas-6 rings rather than one. Making cartwheel assembly dependent on the co-assembly of two rings could dramatically reinforce the tendency to form a ninefold symmetric ring (see main text). In these models, the density of the Ana2 molecules would fit well into the existing density model. (**C**, **D**) In these models, Ana2 forms the same interactions in the same stoichiometry, but the tetramer bridges adjacent paired spoke layers. This model is less attractive as the Ana2 molecules span two of the basic building blocks of the cartwheel (rather than contributing to the stability of the basic building block), and the Ana2 molecules would not fit into the existing density models (although this is possible if they are held in a flexible manner, so their density is not detectable by EM). In these latter models we show Cep135 tethering the SAS-6 spoke to the pinhead (*Lin et al., 2013*). This interaction likely also occurs in the models shown in (**A**, **B**) (although, for ease of presentation, it is not shown); in the models shown in (**C**, **D**) this putative interaction between Sas-6 and Cep135 would function as the key interaction stabilising the two ring structure. Models where the Ana2 tetramer interacts with Sas-6 molecules in only one layer of the Sas-6 ring structure are also possible, but are not illustrated here.

residue. In vitro mRNA was synthesised from linearised (AscI) pRNA vectors using an mMESSAGE mMACHINE T3 Transcription Kit (Life Technologies), and purified using an RNeasy MinElute kit (Qiagen, Hilden, Germany).

The molar concentration of RNA was normalised according to overall RNA yield, and the theoretical length of each transcript. Embryos expressing RFP-Cnn (*Conduit et al., 2010*) were injected and incubated at 25°C for 90–120 min to allow the mRNA to be translated. Images were acquired using a Perkin Elmer ERS spinning disk system (Volocity software) mounted on a Zeiss Axiovert microscope using a 63× 1.4 NA oil immersion objective and an Orca ER CCD

camera (Hamamatsu Photonics, Japan). Images were processed using either Volocity (Perkin Elmer, USA) or Fiji software (*Schindelin et al., 2012*). Images of injected embryos were classified into different catagories based on a qualitative assessment of the ability of the injected fusion-protein to localize to centrosomes. This was performed blind after random-isation of the images.

### *Drosophila* lines

Flies were kept at 25°C, OregonR and $w^{67}$ (Bloomington Stock Centre) served as wild-type controls. The following mutant alleles and stocks were used in this study: $ana2^{169}$, $ana2^{719}$ (*Wang et al., 2011*), $Sas$-$6^{c02901}$ (*Peel et al., 2007*), Ubq-Ana2-GFP (*Cottee et al., 2013*), Ubq-Ana2-CCA-GFP (this study), WT GFP-Sas-6 (this study) and GFP-Sas-6-F143D (this study). All transgenes were generated by standard P-element mediated transformation (performed by either the Genetics Department, University of Cambridge, UK or Bestgene Inc., Chino Hills, CA), and all fusion proteins are expressed from the ubiquitin promoter, which drives moderate expression in all cell types (*Lee et al., 1988*). GFP-tagged full length Sas-6 was generated by cloning the full length cDNA into the pUbq-GFP(NT) destination vector using the Gateway System (Life Technologies). Point mutations were introduced into full-length *ana2* and *Sas-6* cDNA using site-directed mutagenesis (QuickChange II XL, Agilent Technologies).

### Immunohistochemistry of larval brains, adult testes and early embryos

Brains were dissected, squashed and stained as previously described (*Stevens et al., 2009*). Adult testes were dissected and fixed as described (*Dix and Raff, 2007*). Testes were then incubated with primary antibodies overnight at 4°C followed by washes with PBT and secondary antibody incubation for 4 hr at RT. Slides were washed in PBT and mounted for analysis. Embryos from 0–2 hr egg collections were aged for 1 hr at 25°C and were fixed and stained as previously described (*Stevens et al., 2009*). To preserve the GFP signal in embryos expressing either WT GFP-Sas-6 or GFP-Sas-6-F143D, embryos were fixed in 14.4% microfiltered FA solution containing 100 mM PIPES (pH 7.0), 2 mM EGTA and 1 mM $MgSO_4$ for 5 min. The following antibodies were used: sheep anti-Cnn (1:1000) (*Cottee et al., 2013*), guinea pig anti-Asl (1:500) (*Cottee et al., 2013*); GFP-Booster (ChromoTek, Germany) was used at 1/500 to enhance the GFP signal. Secondary antibodies conjugated to either Alexa Fluor 488 or Alexa Fluor 568 (Life Technologies) were used 1:1000. Hoechst33258 (Life Technologies) was used to visualise DNA.

### Centrosome and centriole quantification

Centrosomes were counted on a Zeiss Axioskop 2 microscope using a 63× 1.25 NA objective. Images were acquired in Metamorph (Molecular Devices, UK) using a CoolSNAP HQ camera (Photometrics, Tucson, AZ) and processed using Fiji (*Schindelin et al., 2012*) and Inkscape (www.inkscape.org/) for image assembly. Only brain cells in metaphase were scored (based on DNA morphology), and only centrosomes that clearly stained for both Asl and Cnn were counted. A total of at least 300 cells from at least five brains were analysed for each genotype. Centriole length was measured in fixed meiosis II spermatocytes using the line drawing and measuring tool in Fiji. Length in pixels was converted into µm. At least 30 testes were analysed for each genotype. Centrioles were also examined in living testes dissected in PBS. Testes were transferred to a coverslip with a drop of saline buffer and gently squashed between the coverslip and slide and imaged on the Zeiss Axioskop 2 system described above.

### Western blot analysis

The following primary and secondary antibodies were used: rabbit anti-Ana2 (3:500), (*Stevens et al., 2010a*), rabbit anti-Sas-6 (1:500) (*Basto et al., 2006*), mouse anti-GFP (1:500, Roche, Switzerland), mouse anti-actin (1:1000, Sigma-Aldrich, St. Louis, MO), anti-mouse HRP (1:3000, GE Healthcare, UK) and anti-rabbit HRP (1:3000, GE Healthcare).

### Recombinant protein expression and purification

The cDNA sequences encoding *Drosophila melanogaster* $Ana2^{193–229}$ (CCCD) was inserted into a custom 'pLip' vector, which encodes two, TEV protease cleavable, His-tagged lipoyl domains (from *Bacillus stearothermophilus* dihydrolipoamide acetyltransferase), one fused at either terminus of the insert (*Cottee et al., 2013*). We term the resulting peptide-fusion a 'diLipoyl fusion protein'. CCCD-A, CCCD-S, and CCCD-D variants were subcloned from the pRNA plasmids described above.

All constructs were expressed in *Escherichia coli* B834 (DE3) cells in LB broth, and purified using Ni-NTA affinity, and size exclusion chromatography. The Ana2 CCCD was purified from its diLipoyl fusion protein by proteolytic cleavage, size exclusion, and ion exchange chromatography. The construct contains a GGS motif at the N-terminus, and an EFGENLYFQ motif at the C-terminus—remnants of the cloning and protease cleavage sites.

*E. coli* codon-optimised cDNA encoding Human STIL[717–758] (CCCD) was synthesised (Genewiz, South Plainfield, NJ) and inserted into the pLip vector. diLipoyl-STIL[717–758] was expressed in *E. coli* C41 (DE3) and purified as for diLipoyl-Ana2[193–229]. STIL[717–758] alone was purified by proteolytic cleavage, followed by reverse Ni-NTA chromatography, and size exclusion. The construct contains the same remnants of the cloning and protease cleavage sites as described above.

Sas6 fragments were cloned from *D. melanogaster* cDNA (AAL68137) into a pETM-14 (EMBL) vector encoding a cleavable N-terminal His tag. The F143D mutation was inserted using a Quikchange II XL mutagenesis kit (Agilent Technologies, Santa Clara, CA). Sas-6[1–241] (WT) and Sas-6[1–241] (F143D) were expressed in *E. coli* B834 (DE3) and purified using Ni-NTA and SEC chromatography. Sas-6[1–216] (F143D) was similarly expressed, however the His-tag was removed via proteolytic cleavage and reverse Ni-NTA chromatography prior to SEC. Sas-6[1–216] (F143D) contains a GP at the N-terminus, and a G after the initiator methionine, due to the cloning and protease cleavage sites.

## Electron microscopy
Protein samples were diluted to 33.3 µg/ml in water. 30 µl of sample was deposited for 2 min onto a 200 mesh, glow discharged carbon coated copper grid. The sample was negatively stained by applying 2% wt/vol uranyl acetate for 10 s before blotting, and air-drying the grid. Samples were viewed using an FEI Tecnai 12 TEM (FEI, Hillsboro, OR), at 120 kV, 43,000× magnification.

## SEC MALS analysis
Samples were dialysed into 50 mM Tris pH 7.5, 150 mM NaCl, 1 mM DTT. 100 µl of protein sample was injected onto an S200 10/300 column (GE Healthcare). The light scattering and refractive index were respectively measured in-line by Dawn Heleos-II and Optilab rEX instruments (Wyatt Technology, Santa Barbara, CA), as the samples eluted from the column. Data were analysed using ASTRA software (Wyatt Technology) assuming a dn/dc value of 0.186 ml/g.

## Mass spectrometry
Protein samples were desalted with a Chromolith RP-18e column (Merck, Kenilworth, NJ). These samples in Acetonitrile:water + 0.1% Formic acid were introduced by electrospray ionisation into a Micromass LCT Premier XE orthogonal acceleration reflecting TOF mass spectrometer in positive ion mode (Micromass, Milford, MA). The resultant m/z spectra were converted to mass spectra by using the maximum entropy analysis MaxEnt in the MassLynx suite of programs.

## Crystallography
*D. melanogaster* Ana2 CCCD was dialysed into 20 mM Tris pH 7.5, 150 mM NaCl, 1 mM DTT, and concentrated to 41–43 mg/ml. Initial crystals grew readily at 20°C overnight in sitting drops, using the Stura/Macrosol and Morpheus screens (Molecular Dimensions, Newmarket, UK). The best diffracting crystal grew in an optimisation screen, using 160 nl protein solution and 40 nl of mother liquor (100 mM HEPES mix (71% pH 7.2, 29% pH 8.2), 42% PEG 600). Crystals typically grew to their maximal size within 2–4 days and were fished and flash frozen in liquid nitrogen within 1–14 days. PEG 600 in the mother liquor served as cryoprotectant.

*D. melanogaster* Sas-6[1–216] (F143D) was purified in 50 mM Tris pH 7.5, 150 mM NaCl, and concentrated to 41.6 mg/ml. Small rod-like crystals grew after ~3 weeks at 21°C in drops containing 300 nl protein solution, and 100 nl mother liquor (0.1 M Bicine/Trizma mix (pH 8.5), 20% wt/vol PEG 550MME, 10% wt/vol PEG 20 K, 30 mM NaNO$_3$, 30 mM Na$_2$HPO$_4$, 30 mM (NH$_4$)$_2$SO$_4$). Crystals were fished and flashed frozen after ~4 weeks with PEG 550MME in the mother liquor serving as cryoprotectant.

## Crystal data collection and processing
Ana2 CCCD data were collected at Diamond beamline I03. Due to the high resolution of diffraction, a short wavelength (0.7293 Å) was used to maximise the number of reflections collected on the detector. A high (0.8 Å) and low resolution sweep were processed using the Xia2 pipeline

(*Winter, 2009*), using XDS (*Kabsch, 2010*) and AIMLESS (*Evans and Murshudov, 2013*). Processing statistics suggest that, given a more optimal beamline setup, useful data could be collected to a higher resolution than 0.80 Å. The structure was solved via molecular replacement (Molrep) using a helix (chain A, residues 2–31) from PDB entry 1UO4 (*Yadav et al., 2005*)—an engineered coiled coil peptide. We retrospectively found that the Ana2 CCCD could be trivially solved via direct methods, using ACORN (*Jia-xing et al., 2005*). Autobuilding was carried out using Buccaneer (*Cowtan, 2006*). Refinement was carried out in Phenix.refine (*Afonine et al., 2012*) and Refmac (*Murshudov et al., 2011*), using anisotropic B factor refinement and hydrogens modelled in riding positions. Manual rebuilding was performed in Coot (*Emsley and Cowtan, 2004*).

Sas-6$^{1–216}$ (F143D) data were collected at the ESRF beamline ID23-2. Data were processed using the Xia2 pipeline (*Winter 2009*), using XDS (*Kabsch, 2010*) and AIMLESS (*Evans and Murshudov, 2013*). Phasing was carried out by molecular replacement in Phaser (*McCoy et al., 2007*) using an ensemble of monomeric SAS-6 structures (2Y3V (A/B/D), 2Y3W (A/B) 3Q0X (A/B)) (*Kitagawa et al., 2011*; *van Breugel et al., 2011*) prepared for MR using Chainsaw to trim sidechains to the last common atom (*Stein, 2008*). Autobuilding was initially carried out using Buccaneer (*Cowtan, 2006*) to build into maps that had been solvent flattened using Parrot (*Cowtan, 2010*). Density for the C-terminal part of the CC is weak, and only continuous at lower map contours (~0.6 σ) and was initially built with the aid of a solvent mask in autoBUSTER (*Bricogne et al., 2011*). Refinement and model building were carried out using autoBUSTER and Phenix.refine (*Afonine et al., 2012*) with model building carried out in Coot (*Emsley and Cowtan, 2004*).

## Acknowledgements

We thank Diamond Light Source for beamtime (proposal mx9306) and the staff of beamlines I04-1 and I03, and the European Synchrotron Radiation Facility for beamtime (proposal mx1305) and the staff of beamline ID23-2. We thank Pietro Roversi for help with X-ray data collection, Alan Wainman for help with fly husbandry, Filip Cvetko for help with mapping the Ana2 transgenes and Errin Johnson and the Dunn School EM facility for help with EM sample preparation and imaging. We also thank David Staunton and the Oxford Biochemistry Department Biophysical suite for conducting the mass spectrometry and help with additional unpublished experiments. The structures presented in this study have been deposited in the PDB under the codes: 5AL6 (Ana2 CCCD) and 5AL7 (Sas-6). This work was supported by a BBSRC studentship (MAC) and Wellcome Trust Senior Investigator Awards to JWR and SML (104575/Z/14/Z, and 100298/Z/12/Z, respectively) (NM, SJ, JL, JWR and SML).

## Additional information

### Funding

| Funder | Grant reference | Author |
| --- | --- | --- |
| Wellcome Trust | 104575/Z/14/Z | Jordan W Raff |
| Wellcome Trust | 100298/Z/12/Z | Susan M Lea |
| Biotechnology and Biological Sciences Research Council (BBSRC) | studentship | Matthew A Cottee |
| Medical Research Council (MRC) | G0900888 | Steven Johnson |

The funders had no role in study design, data collection and interpretation, or the decision to submit the work for publication.

### Author contributions

MAC, Conception and design, Purified protein constructs, Carried out biophysical analyses, Crystallised and solved the Ana2 CCCD and Sas-6 structures, Cloned and purified RNA constructs, Cloned Ana2 and Sas-6 constructs for generation of transgenic fly lines, Contributed to the writing of the manuscript; NM, Conception and design, Carried out the Drosophila in vivo work, Contributed essential unpublished data, Contributed to the writing of the manuscript; SJ, Crystallised and solved the Sas-6 structure, Experimental design, Carried out biophysical analyses, Drafting or revising the article; JL, Cloned and purified RNA constructs, Contributed the RNA injection experiments, Drafting

or revising the article; JWR, Conception and design, Contributed to the writing of the manuscript, Analysis and interpretation of data; SML, Conception and design, Crystallised and solved the Ana2 CCCD and Sas-6 structures, Contributed to the writing of the manuscript

## Additional files

### Major datasets

The following datasets were generated:

| Author(s) | Year | Dataset title | Dataset ID and/or URL | Database, license, and accessibility information |
|---|---|---|---|---|
| Cottee MA, Lea SM | 2015 | Central Coiled-Coil Domain (CCCD) of Drosophila melanogaster Ana2. a natural, parallel, tetrameric coiled-coil bundle | http://www.rcsb.org/pdb/explore/explore.do?structureId=5AL6 | Publicly available at RCSB Protein Data Bank (Accession No: 5AL6). |
| Cottee MA, Johnson S, Lea SM | 2015 | N-terminal fragment of Drosophila melanogaster Sas-6 (F143D), dimerised via the coiled-coil domain | http://www.rcsb.org/pdb/explore/explore.do?structureId=5Al7 | Publicly available at RCSB Protein Data Bank (Accession No: 5AL7). |

The following previously published datasets were used:

| Author(s) | Year | Dataset title | Dataset ID and/or URL | Database, license, and accessibility information |
|---|---|---|---|---|
| Yadav MK, Redman JE, Leman LJ, Alvarez-Gutierrez JM, Zhang Y, Stout CD, Ghadiri MR | 2005 | Structure based engineering of internal molecular surfaces of four helix bundles | http://www.rcsb.org/pdb/explore/explore.do?structureId=1UO4 | Publicly available at RCSB Protein Data Bank (Accession No: 1UO4). |
| Van Breugel M | 2011 | N-TERMINAL HEAD DOMAIN OF DANIO RERIO SAS-6 | http://www.rcsb.org/pdb/explore/explore.do?structureId=2Y3V | Publicly available at RCSB Protein Data Bank (Accession No: 2Y3V). |
| Van Breugel M | 2011 | N-TERMINAL HEAD DOMAIN AND BEGINNING OF COILED COIL DOMAIN OF DANIO RERIO SAS-6 | http://www.rcsb.org/pdb/explore/explore.do?structureId=2Y3W | Publicly available at RCSB Protein Data Bank (Accession No: 2Y3W). |
| Kitagawa D, Vakonakis I, Olieric N, Hilbert M, Keller D, Olieric V, Bortfeld M, Erat MC, Flueckiger I, Gonczy P, Steinmetz MO | 2011 | N-terminal coiled-coil dimer domain of C. reinhardtii SAS-6 homolog Bld12p | http://www.rcsb.org/pdb/explore/explore.do?structureId=3Q0X | Publicly available at RCSB Protein Data Bank (Accession No: 3Q0X). |

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
