## [Decision Letter]

Thank you for submitting your work entitled “The homo-oligomerisation of both Sas-6 and Ana2 is required for efficient centriole assembly in flies” for peer review at *eLife*.

We are happy to say that your paper has been favourably reviewed, and we have only a few minor comments, which we are sure you will be able to answer.

Sas-6 and Ana2 are core centriolar components in flies and homologues of these are also important for centriolar assembly in other organisms. In this work, the authors investigate how these proteins contribute to centriolar structure and function. They accomplish this mainly by solving the structures of critical domains of these two proteins, the coiled-coil domain of Ana2 and the N-terminal domain of Sas-6.

Although the Ana2/STIL proteins are rather poorly conserved overall, the coiled coil domain is a conserved feature and the authors show that this region is required to target Ana2 to centrioles. This region alone is able to form stable tetramers in solution and, accordingly, a crystal structure revealed a parallel, symmetrical 4 helix bundle. Taking advantage of this structure, the authors made specific point mutations predicted to prevent tetramerization. This mutant both prevented tetramerization in vitro and, importantly, prevented localization to centrioles in vivo.

Sas-6 proteins are more conserved and it had previously been shown that they need to homo-oligomerize to function in centriole duplication. The C-terminus is important for dimerization and the N-terminus is important for homo-oligomerization. The nature of the oligomerization appears to vary between species and may form rings or spirals depending on species. So to address how Sas-6 behaves, they solved the crystal structure of the N-terminal domain of Sas-6. This domain formed large aggregates. Mutation of conserved residue important for the N–N interaction in multiple species also prevented oligomerization in Sas-6. The structure of this dimer was solved to investigate how Sas6 may oligomerize into the canonical cartwheel structure. This structure was similar to the homologues region in other organisms and therefore also likely oligomerizes into a 9-fold ring. Importantly, the mutation that prevented Sas-6 oligomerization perturbed centriole duplication in vivo.

This work is an excellent example of how high resolution structural data allows one to make specific mutations with predictable and testable outcomes in vivo. Prevention of tetramerization of Ana2 and oligomerization of Sas-6 had very specific effects on centriole assembly. Data such as this is critical to tease out the important similarities and differences between species. We recommend publication and have only a couple of minor comments below.

1) Is there a reason why a Ramachandran Analysis was not included in the refinement statistics for both structures?

2) In Figure 5, it would be helpful to mention what antibody was used to detect the Ana2 constructs. The same goes for Sas-6 in Figure 7.

---

## [Author Response]

*1) Is there a reason why a Ramachandran Analysis was not included in the refinement statistics for both structures*?

While refining each structure, we used Ramachandran analysis to ensure the chemical feasibility of our models. The analysis didn’t highlight any issues with our finalised structures, and all modelled residues fall within allowed regions, with all (Ana2 CCCD), or the vast majority (for SAS-6) in favoured regions of Ramachandran space. We initially did not include this information for brevity, but we have now expanded the crystallographic tables to include this information.

*2) In*
Figure 5*, it would be helpful to mention what antibody was used to detect the Ana2 constructs. The same goes for Sas-6 in*
Figure 7.

We have now more clearly listed all antibodies (along with their relevant manufacturers/citations) including rabbit anti-Ana2 (3:500), (66), and rabbit anti-Sas-6 (1:500) (3) in the Materials and methods. Additionally, we have now added “α” symbols into Figure 5 and 7A to clarify which antibodies were used on each blot.

We have also corrected a few minor mistakes, added some references that were missing and amended the text to take into account some work that was published while our paper was under review (most noticeably the Kratz et al., study looking at the interaction between STIL and Plk4).